# One-photon three-dimensional printed fused silica glass with sub-micron features

Ziyong Li[1,2], Yanwen Jia[1,2,3], Ke Duan[1,2,4], Ran Xiao ®[1,2], Jingyu Qiao[1,2], Shuyu Liang[1,2], Shixiang Wang[5,6], Juzheng Chen[1,2], Hao Wu ®[1,2], Yang Lu ®[2,7] ✉ & Xiewen Wen ®[5,6] ✉

The applications of silica-based glass have evolved alongside human civilization for thousands of years. High-precision manufacturing of three-dimensional (3D) fused silica glass objects is required in various industries, ranging from everyday life to cutting-edge fields. Advanced 3D printing technologies have emerged as a potent tool for fabricating arbitrary glass objects with ultimate freedom and precision. Stereolithography and femtosecond laser direct writing respectively achieved their resolutions of ~50 μm and ~100 nm. However, fabricating glass structures with centimeter dimensions and sub-micron features remains challenging. Presented here, our study effectively bridges the gap through engineering suitable materials and utilizing one-photon micro-stereolithography (OμSL)-based 3D printing, which flexibly creates transparent and high-performance fused silica glass components with complex, 3D sub-micron architectures. Comprehensive characterizations confirm that the final material is stoichiometrically pure silica with high quality, defect-free morphology, and excellent optical properties. Homogeneous volumetric shrinkage further facilitates the smallest voxel, reducing the size from $2.0 \times 2.0 \times 1.0 \ \mu m^3$ to $0.8 \times 0.8 \times 0.5 \ \mu m^3$. This approach can be used to produce fused silica glass components with various 3D geometries featuring sub-micron details and millimetric dimensions. This showcases promising prospects in diverse fields, including micro-optics, microfluidics, mechanical metamaterials, and engineered surfaces.

Transparent fused silica glass is an indispensable material for a vast array of scientific and industrial applications due to its exceptional properties[1,2]. Due to its superior optical transparency, thermal stability and chemical resistance, fused silica glass is the preferred material for numerous high-end applications, including mechanics[3,4], photonics[4–6],

micro-fluidics[7,8], and chemistry applications[8]. Despite its performance advantage, fused silica glass has always been notoriously difficult to manufacture due to its poor processability. Micro and nanodevice manufacturing and rapid prototyping have been hampered by the laborious, expensive, and hazardous nature of silica glass micro-

[1]Department of Mechanical Engineering, City University of Hong Kong, Kowloon, Hong Kong, SAR, China. [2]Nano-Manufacturing Laboratory (NML), Shenzhen Research Institute of City University of Hong Kong, Shenzhen 518057, China. [3]Department of Chemistry, Southern University of Science and Technology, Shenzhen 518055, China. [4]Department of Material Science and Engineering, College of Aerospace Science and Engineering, National University of Defense Technology, Changsha 410073, China. [5]State Key Laboratory of Ultra-precision Machining Technology, Department of Industrial and Systems Engineering, The Hong Kong Polytechnic University, Kowloon, Hong Kong, SAR, China. [6]Research Institute for Advanced Manufacturing, Department of Industrial and Systems Engineering, The Hong Kong Polytechnic University, Kowloon, Hong Kong, SAR, China. [7]Department of Mechanical Engineering, The University of Hong Kong, Pokfulam Road, Hong Kong, SAR, China. ✉e-mail: ylu1@hku.hk; xw.wen@polyu.edu.hk

structuring processes, as well as the difficulty of achieving sophisticated three-dimensional architectures[9,10]. Emerging 3D printing technology, also known as additive manufacturing (AM), uses digital designs and layer-by-layer accumulation to construct complex architectures via serial deposition, thereby streamlining manufacturing processes[11–16]. Current high-resolution glass 3D printing techniques typically involve precise and localized photopolymerization, which solidifies liquid polymer resin into a solid phase. These technologies are principally represented by two mainstreams: stereolithography (SLA)[16] and two-photon lithography (TPL)[17–23]. Through the selective sequential polymerization of glass precursor, SLA and TPL have

respectively evolved as solutions for the fabrication of 3D fused silica glass objects at a resolution of tens of microns[16] and hundreds of nanometers[19,21].

Despite the successful demonstration of constructing 3D architected glasses using macroscale SLA and nanoscale TPL techniques, there are still numerous theoretical and practical obstacles that impede the efficient production of 3D silica objects desiring submicron resolution. Current SLA technologies employed in glass 3D printing suffer from the limitation of minimum producible feature sizes, which are typically on the order of tens of microns. On the other hand, the small field of view of TPL makes large-scale fabrication

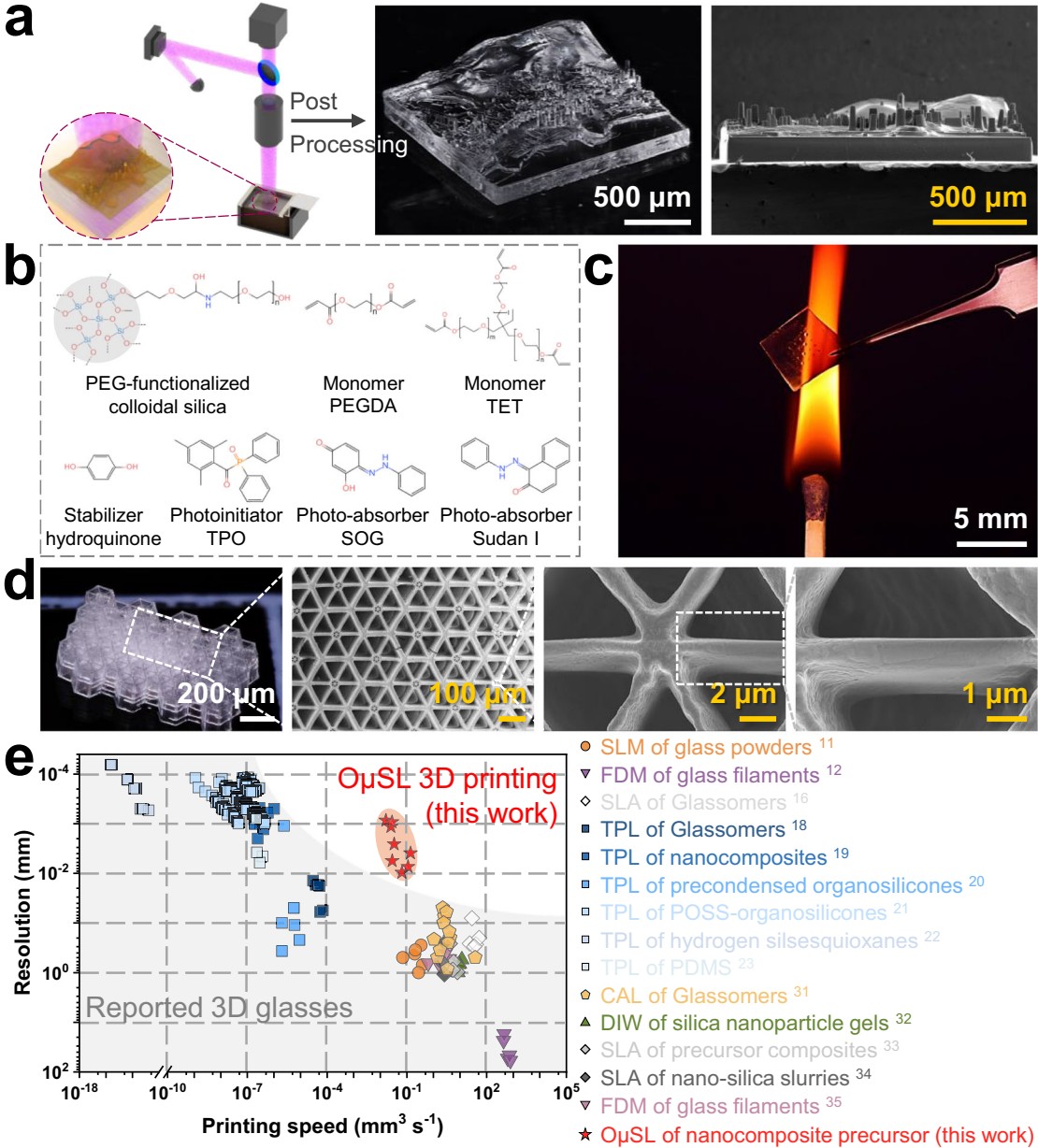

**Fig. 1 | 3D printing transparent fused silica glass with one-photon lithography.**
**a** Illustration of OµSL configuration and its forming process; bottom left inset: schematic of the 3D miniature Hong Kong dioramas structure in printing; and optical & electron microscopic images of the 3D-printed fused silica glass miniature Hong Kong dioramas microstructure. **b** Chemical structures of the nanocomposite precursor. PEG-functionalized colloidal silica as the silica source, PEGDA and TET as the PEG-based bifunctional and trifunctional monomer, hydroquinone as the stabilizer, Irgacure TPO as the photoinitiator, and SOG and Sudan I as the photo-

absorbers. **c** High-temperature stability demonstration of an OµSL 3D-printed transparent fused silica glass MLA at hundreds of degrees. **d** Optical & electron microscopic images of the 4 × 6 honeycomb structure with slender threads centrally suspended. **e** Plot of the finest resolution against the maximum printing speed of the 3D-printed transparent fused silica glass in our work, together with the data of other reported 3D-printed transparent fused silica glass for comparison[11,12,16,18–23,31–35].

especially challenging. It is limited to parts with restricted dimensions, and achieving multi-format stitching presents considerable difficulties. Even though it has been reported recently[20–23] that organosilicon resins are capable of fabricating high-precision glass optics below the glass transition temperature, this technology is still in its early stage, and only a few appropriate organic precursors have been developed so far. The nanocomposite system is still widely considered a more sophisticated solution, and in principle, it is better suited for the fabrication of pure, high-quality fused silica glass for applications in microelectronics and micro-/nano- photonics. In addition, the micro-explosions resulting from higher-order absorption processes[17,24] may damage the printed microstructures and render the entire laborious TPL process ineffective. Lastly, the desired femtosecond laser and the precise instruments are expensive[13]. All the aforementioned limitations have impeded the efficient realization of 3D fused silica glass (sub-)microdevices[25] and their application in micro-optics[4], micro-fluids[7], micro-mechanics[26], or droplet-manipulate micro-surfaces.

Unlike SLA and TPL, one-photon micro-stereolithography (OμSL) is a one-photon lithography technique that permits iterative polymerization in all corresponding regions upon UV light interaction with the photoresist, incorporating both high-resolution and high-speed advantages[27–30]. By positioning a high-precision reduction lens between the projector and the resin tank, the resolution is finely modulated to the desired levels. In addition, OμSL 3D printing is carried out in the top-down direction, which reduces the need for support structures and protects intricate details from damage. Therefore, the OμSL systems can achieve a printing resolution as low as 2 μm and exhibit a dimensional tolerance of up to ± 2 μm. In the prior efforts[16], μSL was utilized with the glass nanocomposites, but only tens-of-micros resolution was demonstrated due to the unevenness of the nanocomposites made from fumed silica nanoparticles and binders. Nonetheless, we propose that with chemically modified monodispersed silica nanocomposites, miscellaneous silica glass 3D objects with sub-micron features and with sizes up to several millimeters without stitching can be easily fabricated. After the OμSL of a silica nanocomposite photopolymerizable precursor, the precursor is transformed into high-quality fused silica glass through a series of post-processing steps. Post-processing can lead to additional homogeneous volumetric shrinkage, which is determined by the initial mass loading of silica in the nanocomposite precursor, allowing for further enhancements in achieving minimum feature size for printed parts. The printed fused silica glass has the same optical transparency as fused silica glass that is commercially available. Moreover, our proposed method effectively overcomes the resolution and efficiency limitations of existing fused silica glass construction techniques, bridging the gap between the macro and nano scales. It produces fused silica glass with miscellaneous 3D (sub-)microarchitectures, prominent build quality, optical clarity, mechanical properties, and chemical resistance. It is a highly promising technique for micro-optics, micro-fluidics, micro-mechanics, and micro-surfaces applications.

## Results

### OμSL 3D printing of (sub-)micron fused silica glass

Our proposed method requires the refinement of a precursor consisting of well-dispersed silica nanoparticles encapsulated in an OμSL-polymerizable monomeric matrix. This precursor must satisfy the following criteria: i) the particle diameter of silica ultrafine powders must be decreased to achieve sub-micron resolution; ii) the refractive indexes of silica nanoparticles and photopolymerizable monomer must be matched to produce a transparent precursor with minimal light extinction and scattering; and iii) the dispersion of silica nanoparticles in monomer must be sufficiently stable and homogeneous to prevent agglomerations. Managing all of the aforementioned requirements concurrently presents a formidable practical challenge.

For instance, although smaller nanoparticles are required for higher resolution, ultrafine powders inevitably increase the viscosity of nanocomposites, making them difficult to physically blend during preparation and self-level during deposition. In addition, the intrinsically high surface energy of ultrafine powders makes them susceptible to clustering in order to achieve a more stable state. The heterogeneity of the nanocomposites restricts their use in the production of structures with minute feature sizes and also results in varying silica packing densities between samples, thereby degrading optical properties and even causing white or opaque appearances. To overcome these obstacles, we employ the similarity-interchangeability theory. First, a solution containing uniformly dispersed polyethylene glycol (PEG)-functionalized silica colloidal nanoparticles is mixed with two small-molecule, PEG main chain-based acrylate monomers with high cross-linking efficiency (Fig. 1b). The solvent is then evaporated from the aforementioned premix, resulting in an increase in silica nanoparticle concentration, a decrease in internal stress during post-processing, and an increase in glass yield. Removal of the solvent would also result in a solution that is clear and transparent (Supplementary Fig. 1a), with approximately 95% optical transmission at the curing wavelength of 405 nm (Supplementary Fig. 1b). The size distribution of the silica nanoparticles is exceptionally narrow, ranging from 10 to 60 nm with an average diameter of approximately 20 nm (Supplementary Fig. 2). Utilizing sub-wavelength silica nanoparticles can significantly reduce the ultraviolet (UV) light scattering effect. The nanocomposite precursor exhibits significant shear thinning behavior with a viscosity of 319.38 mPa·s at a shear rate of 30 s$^{-1}$ and a temperature of 30 °C, which is typical for non-Newtonian fluids (Supplementary Fig. 3). The photoinitiator is used as the trigger for photopolymerization, while photo-absorbers and stabilizers are added to improve resolution of exposure dimensions and sample growth direction (where the x−y building plane serves as the exposure dimension and the z-axis denotes the bottom-up sample growth direction). The nanocomposite precursor is exceptionally stable, with a zeta potential of 54.3 mV on average (Supplementary Fig. 4a). It can be stored at room temperature and ambient condition for months without any signs of agglomeration or sedimentation, and without a change in its photocuring properties (Supplementary Fig. 4b).

After successfully developing the nanocomposite precursor, we employed it in OμSL to create 3D structures with sub-micron resolution, as shown in Fig. 1a. Benefiting from the compatibility and uniformity of its components, our precursor enables efficient UV light penetration for selective photopolymerization; the refractive indices of both nanoparticles and monomers are matched, resulting in exceptional transparency and effectively mitigating of light extinction and side scattering; a lower exposure threshold of 0.22 mJ mm$^{-2}$ is achieved, which is a significant improvement compared to the previous μSL effort[16], ensuring expeditious polymerization of the precursor even under a feeble exposure intensities; the excellent flowability (Supplementary Fig. 3) facilitates the self-leveling of the liquid, allowing for serial and rapid formation of 3D microstructures; the photo-absorbers decrease the absorption coefficients of precursor (Supplementary Fig. 5a), preventing shape errors caused by overexposure and obtaining a uniform thickness of approximately 2 μm (Supplementary Fig. 5b) and approximately 800 nm after the final-sintering (Supplementary Fig. 6c) in the sample growth direction (z-axis). By precisely adjusting the input exposure energy and slice thickness (with a minimum of 1 μm for the printing system), it is possible to further reduce the thickness of a printed fused silica glass monolayer to approximately 500 nm after post-processing (Supplementary Fig. 11e). The development of elaborate transparent nanocomposite precursor, tailored with high compatibility, great uniformity, desired rheological and photo-curing properties, which has not been presented in prior studies[16], is the fundamental element in realizing the critical printing resolution of our OμSL system (2 μm).

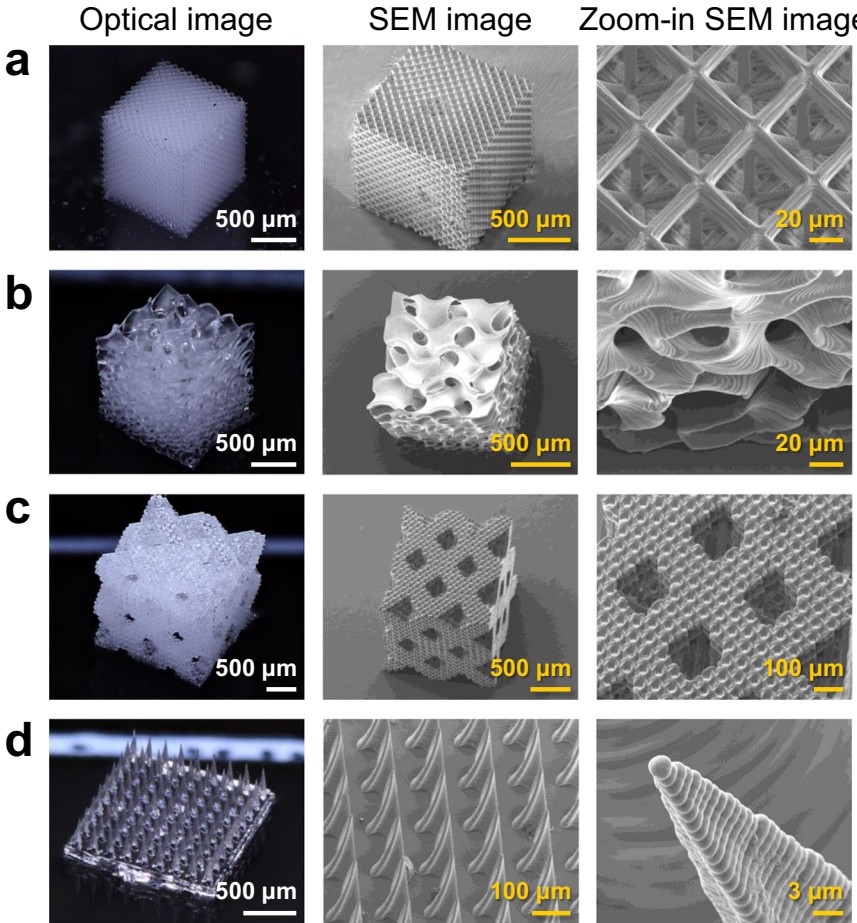

| Optical image | SEM image | Zoom-in SEM image |

**Fig. 2 | Micro-architectures of transparent fused silica glass printed via proposed OμSL technology.** Optical & electron microscopic images of a 3D-printed 12 × 12 × 12 octet-truss lattice structure (**a**), **b** a unit grading gyroid structure, **c** a FCC hybrid hierarchical lattice structure, and **d** a 9 × 9 Snake fang-inspired microneedle arrays; the tip radius is 2.97 μm.

The printed part is then immersed in a 1-methoxy-2-propanol acetate (PGMEA) solvent for development before being rinsed with isopropyl alcohol (IPA). The nanocomposite precursor is post-cured using an ultraviolet light-emitting diode lamp to ensure complete crosslinking, while critical point drying is optional to prevent fine structures from collapsing due to surface tensions and capillary forces.

Following this, the as-printed component is subjected to pyrolysis/debinding, wherein a temperature ramp-up and dwell are employed to facilitate the decomposition of the polymer matrix. The excellent shape retention of the obtained pyrolytic carbon/silica nanoparticle composite can be attributed to the gentle physical shrinkage and mild chemical decomposition during low ramping-up rate pyrolysis/debinding. Subsequently, the as-pyrolyzed portion is subjected to decarbonization/oxidation in an ambient atmosphere to eliminate all residual carbon byproducts, resulting in the aggregation of silica nanoparticles. The silica backbones are converted into dense amorphous fused silica glass during the final sintering steps at 1050 °C under vacuum (Supplementary Fig. 7b). The aforementioned sequential post-processing procedures refine fused silica glasses, resulting in finer features, higher quality, defect-free morphologies, and optical transparency comparable to commercially available fused silica glasses. The combination of a sophisticated precursor, high-resolution printing, and gentle post-heat treatments is considered the cornerstone of our methodology, paving the way for the rapid and straightforward preparation of fused silica glasses with high transparency and intricate micro- and sub-micron geometries.

Figure 1a is an optical and SEM micrograph of the 3D-printed miniature Hong Kong dioramas made from transparent fused silica glass (with optical setup sketch OμSL 3D printing process in Supplementary Fig. 8). As shown in Fig. 1a, the glass dioramas have overall dimensions of 1.50 mm × 1.49 mm × 0.36 mm with a minimum feature size of 1.14 μm (Supplementary Fig. 9). Moreover, the OμSL 3D-printed fused silica glass exhibits exceptional transparency even when subjected to thermal shocks of several hundred degrees, as shown in Fig. 1c. The printed fused silica glass micro-lens array (MLA) sample remained intact following instant heating and subsequent cooling, while the organic glasses are unable to withstand such high temperature, resulting in either softening or combustion. As depicted in Fig. 1d, we construct a 4 × 6 honeycomb structure with fine threads suspended in the center to achieve maximum resolution. The results demonstrate that the proposed technology is capable of sub-micron resolution, as the smallest structure achieved a width of approximately 900 nm (Supplementary Fig. 11c). The resolution was further confirmed by constructing an additional 15 × 10 column array with line arrays affixed on top (Supplementary Fig. 10a–c). The line array comprises four lines in both vertical and horizontal orientations, respectively arranged with varying horizontal spacing and vertical widths. The dimensions of both line width and spacing successively increase from 2 μm (equivalent to one pixel) to 8 μm (equivalent to four pixels), respectively. The proposed technology enables simultaneous sub-micron resolution in both width and distance (Supplementary Fig. 10d), with features exhibiting a line spacing ranging from approximately 800 nm to 4 μm

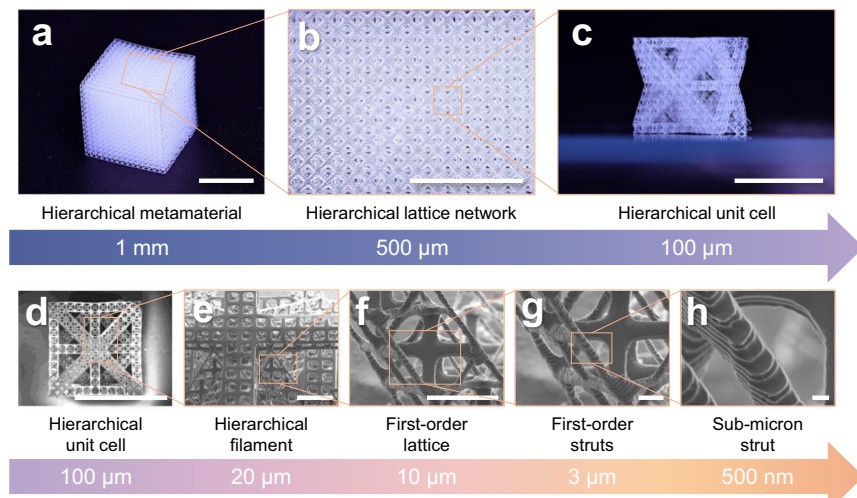

**Fig. 3 | OμSL 3D-printed fused silica glass hierarchical lattice structure with multi-scale critical features.** Optical microscope images of the OμSL 3D-printed fused silica glass hierarchical (**a**) lattice structure, **b** lattice network and **c** unit cell. **d**–**h** Electron microscope images depicting features of structurally hierarchical fused silica glass lattice unit cell shown in (**c**) down to sub-micron in strut size.

(Supplementary Fig. 11a) as well as a line width ranging from approximately 800 nm to 3 μm (Supplementary Fig. 11b). Figure 1e and Supplementary Table 1 compare our proposed OμSL of fused silica glass to other printed fused silica glasses to demonstrate its superior performance in terms of finest feature size and printing speed[11,12,16,18–23,31–35]. The other existing techniques, represented by selective laser melting (SLM)[11], fused deposition modeling (FDM)[12,35], stereolithography (SLA)[16,33,34], computed axial lithography (CAL)[31], and direct ink writing (DIW)[32], possess relatively worse resolution. On the contrary, achieving high-precision fabrication, such as two-photon lithography (TPL)[18–23], always involves navigating challenges related to efficiency and scalability. Our proposed OμSL technology opens up possibilities for fabricating fused silica glass objects with complex geometries and sub-micron features. This proposed technology offers a crucial spatial resolution of approximately 800 nm, effectively bridging the gap between existing macroscale SLA and nanoscale TPL. It surpasses the reported spatial resolution for other 3D printed fused silica glass to date by at least one order of magnitude, and even exceeds that achieved with TPL technique with another type of nanocomposite by tens of times[18,20]. Furthermore, this technology offers a construction speed that is seven orders of magnitude faster and a building area that is two orders of magnitude larger than our previous work obtained by TPL[19], as well as those reported recently[21–23].

To demonstrate the robustness of our proposed techniques, typical optical and SEM images of diverse 3D-printed fused silica glass structures are illustrated in Fig. 2. All these OμSL 3D-printed structures have undergone complete post-processing and final sintering treatment. These optical and SEM images suggest that sophisticated structures with resolutions ranging from sub-micron to microns can be achieved employing the method described above. In particular, a 12 × 12 × 12 octet-truss lattice structure (Fig. 2a) of a 3.78 μm beam width and a near 1 × 1 × 1 mm³ overall dimension, and a unit grading gyroid lattice structure (Fig. 2b) with a similar overall dimension composed of triply periodic minimal surface features of an around 4.71 μm thickness are highlighted to demonstrate the prominent feasibility of the OμSL strategy. Other elaborate structures such as a face centered cubic (FCC) hybrid hierarchical lattice structure (Fig. 2c) with a beam diameter of 3.55 μm and a 9 × 9 Snake fang-inspired microneedle arrays (Fig. 2d) with sharp tips of 2.93 μm are also produced with success.

Additionally, we demonstrate an OμSL 3D-printed fused silica glass hierarchical lattice structure (Fig. 3), which exhibits feature size spanning nearly 5 orders of magnitude, ranging from millimetric scale

(Fig. 3a) to sub-micron scale (Fig. 3h and Supplementary Fig. 11d). The multiscale structure establishes a hierarchical connection from millimetric architectures to sub-micron features, progressively reducing its feature size by several times at each level. Our proposed technique enables the creation of 3D fused silica glass with simultaneous macroscale architectures (Fig. 3a) and sub-micron-scale features (Fig. 3h and Supplementary Fig. 11d), a combination that has not been reported in previous 3D-printed fused silica glass[11,12,16,18–23,31–35] and can be barely achieved using other lithography-based 3D printing techniques[16,18–23,31,33,34].

Following the high-precision OμSL of miscellaneous 3D micro-/nano- architectures, sintering is another most important step in obtaining high-quality fused silica glass components. Optical and SEM observations of the 3D-printed monoliths are compared with those of the as-decarbonized part and the as-sintered part at various temperatures (Supplementary Fig. 12) in order to investigate the shrinkage, deformation, and transparency resulting from sintering, as these parameters are essential for achieving the desired structure and for subsequent application. A comparison between the highly transparent fused silica glass monolith sintered at 1050 °C (Supplementary Fig. 12b) and its as-decarbonized counterpart (Supplementary Fig. 12a) reveals a homogeneously linear shrinkage of 52% and 35%, respectively. The SEM image of the fracture surface of the as-sintered fused silica glass is depicted in the right part of Supplementary Fig. 12b, and a free of defects and pores microstructure can be obtained in the glass matrix at this temperature.

The material analysis measurements were conducted using the printed fused silica glass monoliths with a diameter of 5.0 mm and a thickness 0.5 mm. Additionally, the commercially available fused silica glass monoliths (JGS-1 Grade, very high content of pure silica ($SiO_2 \geq 99.9999\%$), DSP Surface Finished, TTV < 20 μm, Bow/Warp <60 μm, Top Side Ra <1 nm) of identical dimensions were procured from Original Crystal Electronic Technology Co. Ltd (Shandong, China) as the control group. The cross-sectional measurement of the printed fused silica glass was conducted at a location 250 μm beneath the surface, from the center plane of the 0.5-mm-thick monolith. The SEM images in Supplementary Fig. 13b, c depict the printed fused silica glass exhibits an initial defect-free and uniform microstructure, both on the surface and in the cross-section, which is comparable to that of the commercial reference sample (Supplementary Fig. 13a). And the presented energy-dispersive spectroscopy (EDS) mappings further reveal the homogeneous distributions of Si and O elements across all

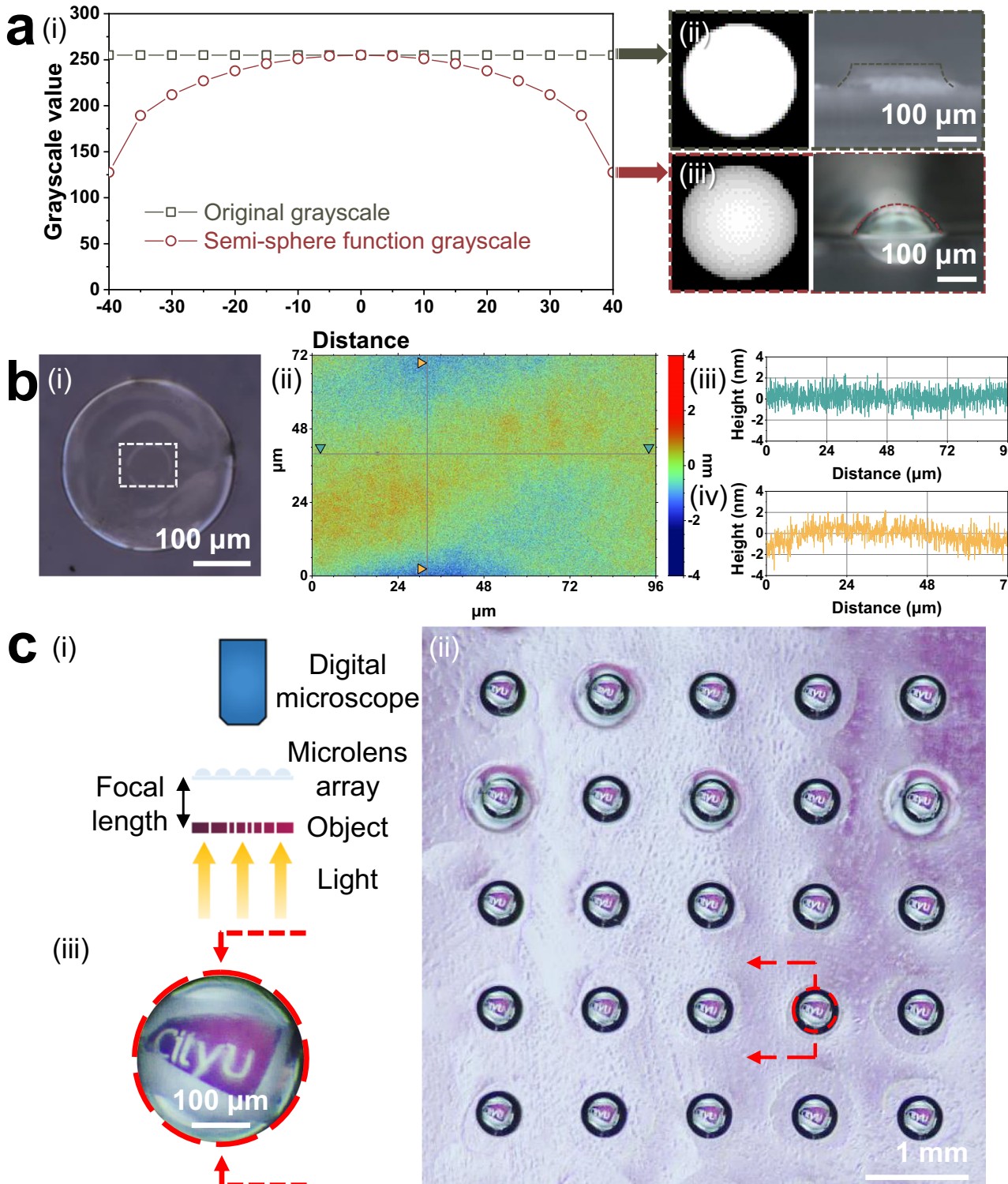

**Fig. 4 | Design, fabrication, characterization, and demonstration of the OµSL 3D-printed fused silica glass MLA. a** Numerical construction and topographical characterization of microlens profiles: (i) numerically designed greyscale distributions, and corresponding greyscale maps and printed microlens profiles (ii) without and (iii) with hemispherical distribution. **b** (i) Optical microscopic image of a single OµSL 3D-printed microlens, and corresponding (ii) white-light interferometer (dotted area within (i)) image with height variations in (iii) X- and (iv) Y-directions (marked by the grey lines). **c** Characterization of the 5 × 5 fused silica glass MLA: (i) schematic illustration of the optical imaging setup, (ii) arrays of the forming miniaturized CityU badge and (iii) corresponding zoomed-up image.

three groups. A high level of consistency in binding energy, intensity, and atomic percentages of all elements were detected within the three groups (Supplementary Fig. 14a–l and Table 2). Despite the well-known inherent limitations of EDS in qualifying light elements[36–41], these findings demonstrate that the quality and purity of the final printed products are well comparable to those of commercially high-quality fused silica glass.

The adventitious carbon (C1$s$ state, C-C bond with a binding energy of 284.8 eV) was employed for XPS calibration, which is an indispensable procedure prior to the analysis of electrically insulating fused silica glass. The printed fused silica glass is composed of pure Si and O elements, as confirmed by the full X-ray photoelectron spectroscopy (XPS) analysis (0–1350 eV, Supplementary Fig. 15a), which aligns with the characteristics observed in the commercially available reference sample. The same normalizations were applied to each characteristic peak (526–536 eV for O1$s$, 97–107 eV for Si2$p$, and 0–35 eV for O2$s$) of both the printed and commercial samples, the fine spectroscopy results (inset of Supplementary Fig. 15a) revealed that both the printed and commercial samples displayed comparable photoelectron binding energy and relative intensity. The peaks for the printed fused silica glass closely matched commercially stochiometric fused silica glass (531.0 ± 0.5 eV for O1$s$ peak, 101.7 ± 0.5 eV for Si2$p$ peak, and 23.6 ± 0.5 eV for O2$s$ peak), indicating their identical chemical state and structural composition (Supplementary Table 3).

To investigate the optical capabilities of OµSL 3D-printed fused silica glass, we measure the ultraviolet-visible (UV-vis) transmission spectrum of a 500-µm-thick printed monolith with a measured range spanning from 300 to 1150 nm (Supplementary Fig. 16). The spectral analysis indicates that the 3D-printed fused silica glass material exhibits exceptional optical transmission properties, achieving an impressive result of approximately 95% across the entire spectrum without any visible absorption peaks.

Using X-ray diffraction, the crystallographic phase of as-sintered silica is characterized (XRD, Supplementary Fig. 15b). In this test, the printed fused silica glass monolith was carefully polished on both sides to mitigate the impact of surface roughness[42,43]. After sintering at 1050 °C, the composite component transformed into amorphous fused silica glass with a single broad peak at 22 degrees. The Raman spectroscopy measurement was performed to characterize the material's structural properties (Supplementary Fig. 15c). In the context, the $\omega_1$ bands (cm$^{-1}$) and $\omega_3$ bands (cm$^{-1}$) relate to bending vibration of the Si(O$_{1/2}$)$_4$ tetrahedrons' Si-O-Si bridges, and $\omega_4$ bands (cm$^{-1}$) can be ascribed to the stretching motion of the Si-O bonds. The $D_1$ bands (cm$^{-1}$) and $D_2$ bands (cm$^{-1}$) correspond to the symmetric stretching of silicon-oxygen ring molecules. The spectrum exhibits no additional peaks, confirming the absence of any residual impurities in the printed fused silica glass. The excellent consistency observed in the spectrum of the printed fused silica glass, when compared to that of the commercial reference sample, confirms its composition as pure silicon dioxide, thus aligning with the composition the commercial fused silica glass.

The transmission electron microscopy (TEM) images in Supplementary Fig. 17a, b demonstrate the dense nature of the printed fused silica glass structures, exhibiting an absence of discernible pores or cracks. Supplementary Fig. 17c–e presents energy-dispersive spectroscopy mapping, which reveals the homogeneous distributions of Si and O elements. The diffraction patterns (inset of Supplementary Fig. 17b) corresponding to the amorphous phase exhibit excellent agreement with the X-ray diffraction analysis. According to Supplementary Fig. 18, the atomic percentage (at %) of silicon was measured to be 33.8 ± 0.1, while that of oxygen was found to be 63.4 ± 0.1 at %. These values closely corresponded to the stoichiometric SiO$_2$. Consequently, the integration of XPS, XRD, Raman spectrum, TEM, and electron diffraction (Supplementary Fig. 15, 17, and 18; Tables 2 and 3) identifies the final materials to be comparable to the stoichiometrically pure silica.

## Micro-lens fabrication and micro-optics applications

Transparent fused silica glass has broad compatibility with micro-optical, microfluidic, and micromechanical applications. However, it is well known that conventional methods for manufacturing silica micro-components cannot produce miscellaneous 3D microstructures. In contrast, high-resolution OµSL enables the production of novel fused silica micro-components, thereby expanding the design, geometry, and resolution degrees of freedom for a variety of high-end fields. The optical imaging capability of the printed fused silica glass microlens arrays (MLA) is demonstrated using the optical microscope system depicted schematically in Fig. 4.

To prevent the staircase effect from degrading the outer texture, a greyscale lithography strategy (Fig. 4a) is used to modify the gradients in exposure intensity for the generation of an optically smooth hemispherical surface. Greyscale images are constructed using data matrices whose elements represent the greyscale values of the pixels. The dimension of the hemisphere data matrix is denoted as $2n+1$, and the greyscale value for each individual pixel is represented by $G(i,j)$. As the height of a conventional convex lens gradually increases from the lens's edge to its center, pixels located at the lens's edge and center should be assigned the minimum and maximum values, respectively. Here, the greyscale follows a hemispherical distribution with a maximum value of 255 and a minimum value of 128 in the printing instrument.

$$G(i,j) = \begin{cases} G_{min} + \frac{G_{max}}{2}\sqrt{1 - (D(i,j)/n)^2} & D(i,j) \leq n \\ 0 & D(i,j) > n \end{cases}, \quad (1)$$

in which $D(i,j) = \sqrt{(n+1-i)^2 + (n+1-j)^2}$ represents the distance from any given pixel to the center. Using a white light interferometer, the surface roughness of a 96 µm by 72 µm region at the center of an OµSL 3D-printed microlens was measured (Fig. 4b). Smooth surfaces were simultaneously achieved in the X and Y directions with a peak-to-valley height variation of 5 nm (Fig. 4b(ii) & (iii)). Further statistical analysis revealed that the microlens could achieve the desired surface roughness of $R_a \approx 0.633$ nm without additional finishing. The roughness outperforms that of any prior studies[16,18,31–33], which can be attributed to the synergistic enhancement of sophisticated precursors and gentle post-processing. The composition of precursor is simple with the organics easily eliminated, resulting in a high-quality fused silica glass comparable to the commercial one; the intricate stepwise post-processing procedures gently manage the microstructures during heat-treatment, thereby facilitating the elimination of defects and enhancement of density; the monodispersed ultrafine silica nanoparticles reduce the diffusion distance of gaseous byproducts, resulting in a uniform final product structure devoid of any pores. As depicted in Fig. 4c(i), the MLA is positioned between a digital microscope objective and an object, such as a 3D-printed skeletonized CityU logo. Light is sequentially transmitted through the object and MLA to form an image that is captured by the digital microscope. As shown in Fig. 4c(ii) and 4c(iii), upon illumination, the microscope captures distortion-free miniaturized images of the CityU logo from 5 × 5 different lenses of MLA, which also possess great uniformity, clear quality, high contrast, and high sharpness. The demonstration demonstrates the immense potential of OµSL 3D-printed fused silica glass for a variety of optical micro-components for cutting-edge solutions in light field regulating, wavefront sensing, medical diagnosing, and optical signal processing.

## Microlattice metamaterials and mechanical characterizations

Superior and desirable performance has been achieved by a greater variety of three-dimensional lattice structures made possible by modern manufacturing techniques. Unfortunately, these cutting-edge mechanical metamaterials can only withstand moderate-to-high

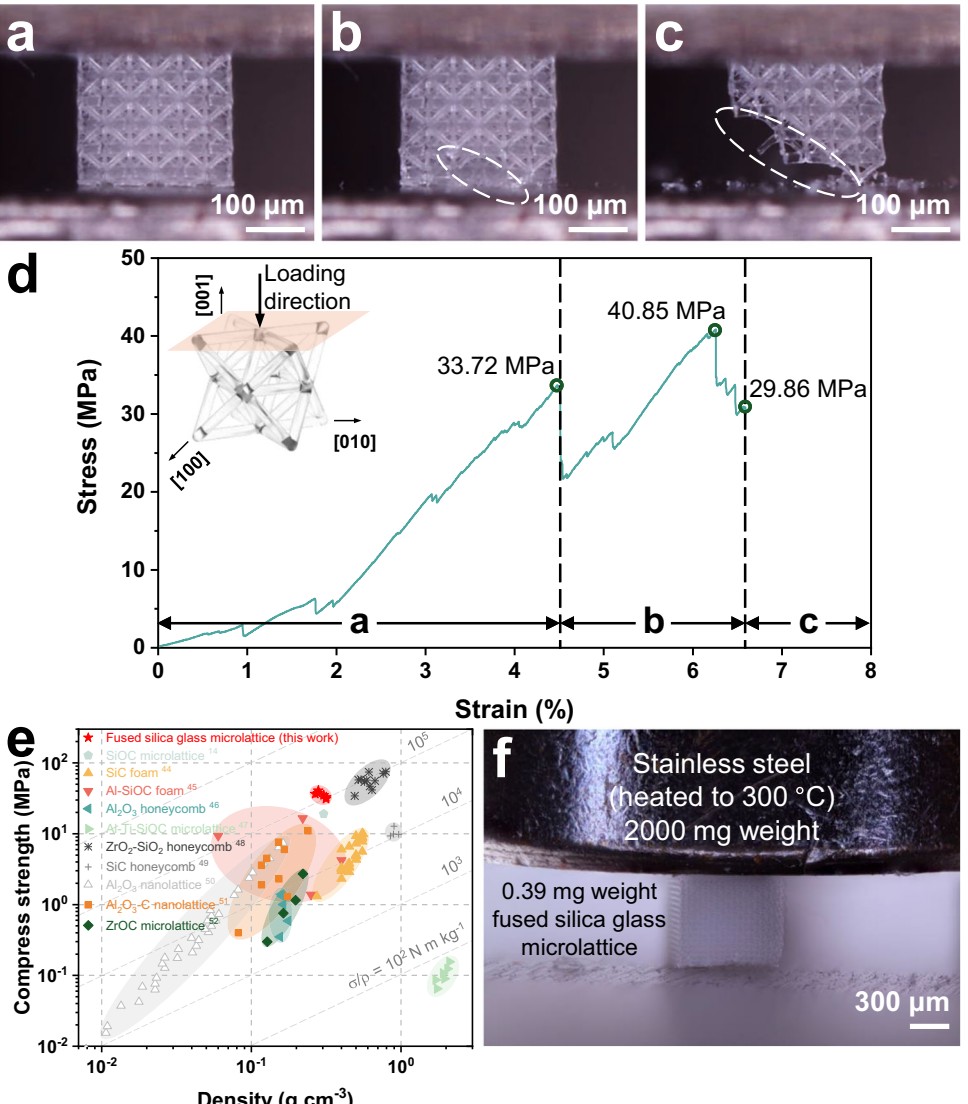

**Fig. 5 | Mechanical characterization of the OµSL 3D-printed fused silica glass microlattice.** In-situ compression images of (**a**) linear elastic region, **b** failure of bottom units marked by dashed circle, and **c** complete failure along the shear band marked by dashed circle. **d** Stress-strain curve of a fused silica glass microlattice. **e** Ashby chart on compressive strength versus density. OµSL 3D-printed fused silica glass microlattices with high specific strength in this work (red star) were compared with other reported high-temperature architected materials[14,44−52]. **f** Demonstration of the highly stable, lightweight, and stiff fused silica glass lattice, a 9 × 9 × 9 octet-truss lattice was lifting a weight 5000 times its own weight at a temperature of 300 °C.

temperatures, high pressure, and harsh chemicals to a limited extent. Even though the integration of fused silica glass and micro lattice architectures can benefit significantly from their ultrahigh stability, ultralight weight, and ultrahigh stiffness, these characteristics are not shared by both materials. To demonstrate the mechanical properties of OµSL 3D-printed fused silica microlattice, a 3 × 3 × 3 octet-truss lattice structure with a beam diameter of 13 µm was fabricated and subjected to an in-situ compression test (Fig. 5a–c and Supplementary Movie 1). During the loading procedure a, the truss exhibited minor linear elastic deformations. At the conclusion of this process, some initial but not complete failure occurred in the bottom units, as depicted in Fig. 5b, which is also indicative of the typical brittle stress-strain failure behavior depicted in Fig. 5d. The component was then loaded to failure, as depicted in Fig. 5c, with major crack propagation along the shear band. In accordance with Fig. 5d, compressive strengths of 33.72 MPa, 40.85 MPa, and 29.86 MPa were attainable at each stage. As depicted in Fig. 5e and Supplementary Table 5, a compressive strength versus density Ashby chart was plotted to facilitate

intuitive comparisons between OµSL 3D-printed fused silica glass microlattices and other high-temperature architected materials[14,44−52]. The specific strength of the fused silica glass microlattices produced by OµSL was $1.22 \times 10^5$ N m kg$^{-1}$, which significantly outperformed other materials of comparable density. This superior performance is attributed to the reduced feature sizes, and according to the weakest link theory, decreasing dimensions can dramatically reduce the risk of failure. And according to the weakest link theory[53,54], and its derived formula:

$$P_f\left(\bar{\sigma}_f\right) = 1 - \exp\left\{-\left(\bar{\sigma}_f/\sigma_0\right)^m \cdot (V/V_0)\right\} \quad (2)$$

$$\bar{\sigma}_f \propto (1/V)^{1/m} \quad (3)$$

in which $P_f\left(\bar{\sigma}_f\right)$ represents the failure risk of the materials subjected to a given stress $\bar{\sigma}_f$; $\sigma_0$ represents the characteristic stress when $P_f\left(\bar{\sigma}_f\right)$ equals to 63.2%; $V$ represents the volume of the materials; $V_0$

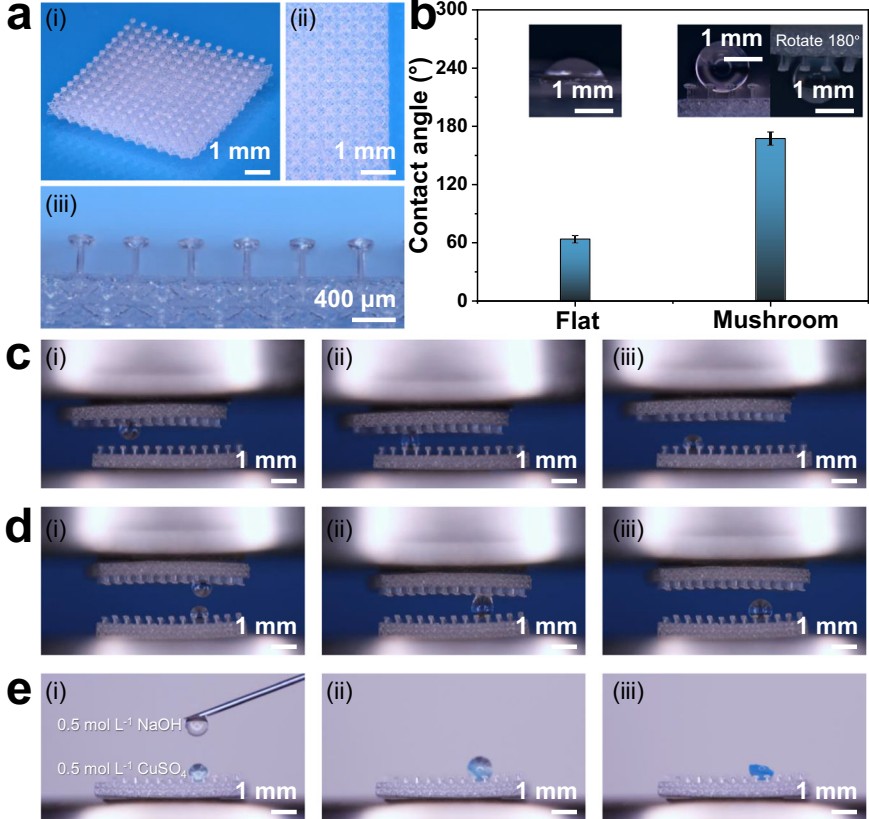

**Fig. 6 | OµSL 3D-printed fused silica glass superhydrophobic micro-surface as well as its characterization and demonstration. a** Morphologies of the fused silica glass micro-surface: (i) global view, (ii) top view and (iii) front view. **b** Contact angles and images of printed fused silica glass micro-surfaces with and without resembling mushrooms; insets: geometries of the droplets on the printed fused silica glass flat- and micro-surface. Controllable operation of individual droplets on the superhydrophobic microsurface of fused silica glass at a high temperature: **c** non-destructive transportation, **d** coalescence, and **e** in-situ chemical reaction.

represent the standardized reference volume; and $m$ represents the Weibull modulus[55].

Therefore, the mechanical characterization suggests that 'the smaller, the stronger' size effects can be fully exploited by OµSL technology, paving the way for the production of lightweight, high-strength fused silica glass mechanical metamaterials with greater reliability than other advanced materials. In the future, our proposed technology will allow for the continued development of novel metamaterials that exploit the inherent high strength of fused silica glass and the failure strain in the presence of minimal defects[1,56].

**Superhydrophobic micro-surfaces and droplets-manipulations**
Inspired by the superhydrophobic surface derived from the "petal effect"[57], Numerous novel superhydrophobic materials have been created and implemented in all facets of society, including aerospace, electronics, and transportation. To date, however, the most developed superhydrophobic surfaces are primarily polymer-based, which suffers from high cost, low durability, poor weatherability, and inadequate chemical resistance. OµSL 3D-printed fused silica glass with a superhydrophobic microsurface can be used to overcome these obstacles. As shown in Fig. 6a, the mushroom-inspired microstructures were regularly centered on the top of octet-truss microlattices rather than the solid substrate in order to be compatible with the shrinkage of the surface textures and substrates. The mushroom had a stem with a diameter $d$ of 66 µm and a height $h$ of 330 µm, a head with a diameter $D$ of 238 µm and a height $H$ of 30 µm, and a gap $G$ that was equal to twice the head diameter, or 476 µm. Despite the printed fused silica glass flat surface suggesting a hydrophilic behavior with a contact angle of 64 °, the OµSL 3D-printed fused silica glass mushroom-inspired microstructures achieved a superhydrophobic behavior with a contact angle of 167 ° (Fig. 6b). In addition, the OµSL 3D-printed fused silica glass microstructures inspired by mushrooms demonstrated excellent droplet control even after a 180-degree inversion, and the droplet adhered to the microstructure without any disturbance (inset of Fig. 6b). The error bars represent the standard deviations, and the measurements were conducted six times on both flat- and micro-surfaces respectively, with average values derived from the test results.

On this basis, a broader range of applications for OµSL fused silica glass superhydrophobic micro-surfaces, such as the droplet operator and microreactor in harsh environments, were further investigated (Fig. 6c-e). In the experiments, the ambient temperature was pre-heated to 80 °C, but the printed fused silica glass micro-surface still exhibited superhydrophobic behavior and a large contact angle. A 3 µL deionized water droplet adhering to the top surface (the inverted fused silica glass micro-surface) was transported without loss to the bottom surface (another upright micro-surface) as depicted in Fig. 6c and Supplementary Movie 2. The droplets coalescence experiment was then repeated with two 2 µL drops of deionized water adhering to both the top and bottom surfaces of the two micro-surfaces. As depicted in Fig. 6d and Supplementary Movie 3, the droplets sequentially contacted, co-dissolved, and coalesced at the bottom of the micro-surface without incurring any loss. These results shed light on the potential of artificial micro-surfaces for specialized applications such as droplet operators, antifouling surfaces, heat exchangers, and water collection systems.

In addition, the superior chemical resistance of as-printed fused silica glass superhydrophobic micro-surfaces permits their use as droplet micro-reactors for aggressive chemicals. To demonstrate the

feasibility, the double displacement reaction between NaOH and CuSO$_4$ was chosen as one of the most classic reactions (Supplementary Movie 4). A 2 μL droplet of CuSO$_4$ was pre-adhered to the microsurface of printed fused silica glass, and a 1 mL syringe was used to deposit a corrosive droplet of NaOH through its needle (Fig. 6e(i)). Throughout the reaction, the droplet maintained its spherical shape and remained standing on the microstructure, resembling a mushroom. This phenomenon results from the well-known reaction CuSO$_4$ + 2NaOH = Cu(OH)$_2$ + Na$_2$SO$_4$ (Fig. 6e(ii)). As the reaction progressed, the agglomeration of Cu(OH)$_2$ precipitated as the solvent-deionized water evaporated at high temperature, leaving a semi-spherical blue product on the microstructure resembling a mushroom (Fig. 6e(iii)). The advantages of the micro-droplet reaction on the OμSL 3D-printed fused silica glass superhydrophobic micro-surface, including high flexibility, adjustable concentration, direct observation, financial efficiency, easy transportation, and resistance to harsh environments, promising novel applications in material synthesis, catalytic processing, biological monitoring, drug release, and medicine screening.

## Discussion

In conclusion, we have developed a method for OμSL 3D printing of a monodispersed silica colloidal nanocomposite precursor that can be converted into high-quality transparent fused silica glass with miscellaneous structures. The precursor is formulated to have superior dispersity, high transparency, low viscosity, and high stability. In conjunction with the appropriate post-processing procedures, it enables the production of sub-micron-resolution 3D fused silica glass components. This process has demonstrated its adaptability in the fabrication of novel sub-micron fused silica glass components, pushing the limits of finer features, more miscellaneous geometry, and greater design freedom for a variety of high-end fields, such as micro-optics, microfluidics, micromechanics, biomedicine, and life sciences.

## Methods

### Materials

Poly (ethylene glycol) diacrylate (PEGDA, average M$_n$ 575), trimethylolpropane ethoxylate triacrylate (TET, average M$_n$ ~ 428), hydroquinone (ReagentPlus, ≥99%), diphenyl (2,4,6-trimethylbenzoyl) phosphine oxide (TPO, 97%), 2,4-dihydroxyazobenzene, 4-(Phenylazo) resorcinol (Sudan Orange G, SOG, Dye content 85%), 1-phenylazo-2-naphthol (Sudan I, Dye content ≥95%), 1-methoxy-2-propanol acetate (PGMEA, ≥96%) and isopropanol (IPA, ACS reagent, ≥99.5%) were all procured by Sigma-Aldrich, Germany. A solution of propylene glycol monomethyl ether containing homogeneously dispersed surface-solubilizing PEG-functionalized colloidal silica nanoparticles (PGM-C-2140Y, SiO$_2$ content 46.8%) was provided by Nissan Chemical Industries Ltd., Japan. One-sided polished silicon wafers were purchased from Hundsun Technologies, China.

### Preparation of nanocomposite precursor

A transparent mixture was firstly prepared by mixing 4280 mg of functionalized silica nanoparticles solution into a premixed monomers of PEGDA (6000 mg) and TET (12000 mg) under magnetic stirring at 500 rpm. Then, 90 mg of hydroquinone was included into the sol for inhibition of the crosslinking of polymerizable monomers. Meanwhile, the mixture was also heated to 150 °C and kept for 180 min to completely remove the solvent PGM. Finally, 180 mg of TPO, 72 mg of Sudan I, and 21.6 mg of SOG were added to the mixture to obtain the nanocomposite photopolymerizable precursor.

### OμSL process

The nanocomposite photopolymerizable precursor was processed by a commercial OμSL system (BMF PμSL nanoArch P130, China). A silicon wafer was firstly ultrasound cleaned in acetone for 600 s, then rinsed three times in IPA to eliminate the pollutants. The as-cleaned silicon substrate was adhesive in the printing platform and then assembled into the BMF PμSL 3D printer. During the AM process, the UV light intensity and exposure time were set to 38.3 mW cm$^{-2}$ and 2.0 s for each single slice, respectively. After finishing this process, the as-printed part along with the silicon substrate, would be immersed in PGMEA for 300 s to dissipate excess precursor and then transferred to IPA to get rid of residual PGMEA. Post-curing was performed using a UV-LED lamp for further ensure complete crosslinking of the printed structure (61 mW cm$^{-2}$, 300 s). Subsequently, the samples in IPA were desiccated through critical point drying to prevent the microstructure from collapsing due to the capillary force or surface tension from the solvent. Finally, the as-dried parts were removed from the substrate with a razor blade, which great care must be taken as the microstructure is delicate and easily broken.

### Heat treatment

To facilitating the removal of crosslinking polymer network and the shape-retention of silica backbones, the pyrolysis/debinding process was undertaken in a tube furnace (Kejing GSL-1700X, China) in a vacuum with a pressure of −0.1 MPa. During this process, the temperature was increased from room temperature to aim temperature with a ramping-up rate of 0.5 °C min$^{-1}$. The dwelling times were respectively set as 120 min and 180 min when temperature reaching 300 °C and 550 °C. After that, the samples were gradually cooled down to room temperature with a ramping-down rate of 2 °C min$^{-1}$. After pyrolysis/debinding, the samples were decarbonized in a box furnace (Kejing KSL-1100X, China) under ambient atmosphere, holding in 600 °C for 120 min with the same heating/cooling rate of 2 °C min$^{-1}$. Finally, the as-decarbonized samples were sintered in a tube furnace (Kejing GSL-1700X, China) with a vacuum pressure of −0.1 MPa. All the ramping-up and -down rates in this process were 2.0 °C min$^{-1}$, and the temperature rise from room temperature to dwelling temperature, which includes 800 °C and 1050 °C with holding time of 120 min and 240 min, respectively.

### Characterizations

Optical transmission was characterized using an ultraviolet-visible spectrometer (UV-vis, Thermo Scientific, Germany). Thermal gravimetric analysis (TGA) was performed on a synchronous thermal analyzer (STA-8000, PerkinElmer, America) in a flowing nitrogen atmosphere where the temperature was gradually raised to a maximum temperature of 600 °C at a rate of 10 °C min$^{-1}$. Particle diameter distribution and zeta potential were examined by a laser particle analyzer (Zetasizer Nano ZS90, Malvern, UK). Deionized water was employed as the solvent for particle size distribution analysis, while ultra-pure water was utilized as the solvent for Zeta potential measurements. The dynamic viscosity of the precursor was measured with a rheometer (Anton Paar MCR 302, Austria). The rheological properties were evaluated by measuring the dynamic viscosities as the functions of shear rate and temperature, using the fixtures with a diameter of 60 mm and geometries of cone/plate and parallel plates, respectively. X-ray photoelectron spectroscopy (XPS) was analyzed using an in-situ X-ray photoelectron spectrometer (PHI 5000 Versa Probe II, ULVAC-PHI, Japan). X-ray diffraction (XRD) analyzes were performed on an X-ray diffractometer (D8 Advance, Bruker, Germany). The microscopic morphologies of each fused silica glass components were studied by a scanning electron microscope (SEM, ZEISS SUPRA 55, Carl Zeiss Corporation, Germany; Phenom Pro, Thermo Fisher Scientific, America; FEI Quanta 450 FEG, Thermo Fisher Scientific, America). The amorphous microstructure of the printed fused silica glass was determined using the field emission transmission electron microscope (FETEM, JEM-3200FS, JEOL Ltd., Japan). The tested TEM sample with a thickness of ~500 nm was carved from the printed fused silica glass monolith via focus ion beam (FIB, Scios, Thermo Fisher

Scientific, America) milling. All the SEM and TEM samples are deposited with Pt coatings using a sputter coater (Leica EM ACE200, Leica Microsystems, Germany) to enhance electrical conductivity, thereby minimizing the charge accumulation during electron microscope observation. All the samples were affixed to the stage using conductive carbon tape during SEM & EDS measurements. Surface roughness was measured using a white light interferometer (Contour GT-K, Bruker, Germany). In-situ compression test of printed fused silica glass microlattices was carried out on a universal testing machine (CMT6503, MTS system, China). The microlattices were subjected to uniaxial compression at room temperature with a specified strain rate of $10^{-2} \, min^{-1}$, using the testing system equipped with a load cell of 100 N. Wetting properties of printed fused silica glass micro-surfaces were investigated by a drop shape analyzer (DSA100S, Kruss Scientific, Germany).

## Data availability

The data generated in this study are provided in the manuscript and Supplementary Information. The data supporting the findings of this study are available from the corresponding authors upon request.

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

## Acknowledgements

Y.L. thanks the Science Technology and Innovation Commission of Shenzhen Municipality under the Shenzhen-Hong Kong-Macau Technology Research Program (Type C, SGDX2020110309300301), Research Grants Council of the Hong Kong Special Administrative Region, China, under the grant RFS2021-1S05 and C7074-23GF, Innovation and Technology Fund of the Hong Kong Special Administrative Region, China under the grant GHP/221/21GD and PRP/054/22 F. X.W. Thanks Research Grants Council of the Hong Kong Special Administrative Region, China under the grant 21206223, and PolyU Startup grant P0048050. X.W. also acknowledges the support from the University Research Facility in 3D Printing (U3DP) and State Key Laboratory of Ultra-precision Machining Technology (SKL-UPMT) of the Hong Kong Polytechnic University.

## Author contributions

Z.L., Y.L., and X.W. conceived the ideas and designed the research; Z.L. performed the precursor preparation; Z.L., Y.J., and K.D. conducted the OμSL printing; Z.L. and Y.J. carried out the sample post-processing; Z.L. developed the heat-treatments; Z.L., Y.J., R.X., J.Q., S.L., S.W., J.C., and H.W. contributed to sample characterization; Z.L. drafted the manuscript; Z.L., Y.L., and X.W. revised the manuscript.

## Competing interests

The authors declare no competing interests.
