## [Peer Review File · Nature Communications]

One-photon Three-dimensional Printed Fused Silica Glass with Sub-micron FeaturesREVIEWER COMMENTS

Reviewer #1 (Remarks to the Author):

This manuscript reports the development of a precursor for additive manufacturing of three-dimensional silica glass objects by stereo-lithography. The main innovation is in the precursor formulation, which improves its optical and rheological properties, enabling the printing of smaller features than previously possible by stereolithography while maintaining benefits such as high printing speeds and large printing volumes. The authors demonstrate their method by printing three-dimensional glass objects of mm size with micrometric features. The new precursor formulation is of significant interest, and the demonstrations in the manuscript advance the state of the art. However, the central claim of this manuscript, and indeed its title, is the 3D printing of fused silica glass with sub-micron features. As detailed below, the authors neither prove that their printed structures consist of fused silica glass nor show convincing sub-micron features, and thus the manuscript is not suitable for publication in its current form.

Major issues:

- Fused silica glass is pure silicon dioxide in amorphous (non-crystalline) form. Thus:
 - o To prove that the 3D printed structures are made of silica, it is necessary to show that the bulk material of the printed structures only contains silicon and oxygen in the stoichiometric ratio 1:2 and to quantify any residual contaminants.

The authors employ two quantitative techniques (EDS and XPS) capable of determining stoichiometry and quantifying contaminants, but their presentation of the results is unclear.

In the case of EDS, the data is provided in Supplementary Data Fig. S9. First, there is no information on the sample. Information should be provided on what size of structure is being measured, if the measurement is done on the printed surface or in a cross-section, and in that case, how close to the printed surface the measurement is made. Furthermore, it should be made clear at what processing stage the sample was measured. Second, a representative EDS point spectrum should be included. In particular, the range from 0 to 2 kV is interesting to determine if any carbon contamination is present. The authors should also report the measured ratios of Si, O, and any detected contaminants, for example, carbon.

In the case of XPS, the data is provided in Supplementary Data Fig S10a. First, the sample information requested above should also be provided here. Second, a broad XPS spectrum (0 – 800 eV) should also be provided, and the measured ratios of Si, O, and any detected contaminants be reported. The authors should also clarify exactly how they conclude that XPS “identifies the final materials as stoichiometrically pure silica” based on their currently presented data. Finally, details should be provided on the “Commercial” reference sample.

- o To prove that the material is amorphous, one must show that it is without long-range order and doesn't contain crystallites. The authors employ XRD to argue that the sample is amorphous. The data is provided in Supplementary Data Fig S10b. First, the sample and reference information requested above is also missing for this measurement and should be provided. Second, the measurement clearly shows a difference between the printed and reference samples. The reference sample has a prominent broad

peak (also known as the amorphous halo) at about 22 degrees (2theta), with a slight hint at another broad peak around twice the angle (a half-order peak). The printed sample also has a prominent broad peak, but it is shifted to about 21 degrees, and the half-order peak is significantly more prominent than in the reference. A shift of the position of the amorphous halo to a lower angle in an X-ray diffraction (XRD) pattern of an amorphous material can indicate an increase in the average atomic spacing in the material and, thus, a less dense amorphous phase. Furthermore, the presence of a half-order peak can indicate a nanostructured material. Considering that the printed structures are made by sintering colloidal silica, the XRD data give reason to believe that the post-processing fails to achieve a homogenous fused silica glass. Indeed, the cross-sectional SEM of the sintered material in Supplementary Data Fig S8b seems to show a residual structure, although the contrast and resolution of that image are too poor to be certain. To convincingly resolve the question of the structure of the printed material, the XRD data should be complemented by transmission electron microscopy and electron diffraction data of a cross-section of a printed structure.

- The claims of patterning “sub-micron features,” as stated in the title of the manuscript, and sub-micron “resolution,” as indicated in Fig. 1 e, are based on the structure shown in Fig. 1d (lower left). The figure shows an isolated line with about 900 nm width in one in-plane dimension. The thickness of the features printed with the method (z-axis resolution) is, however, in all cases above 2 microns. The minimum isolated voxel that can be printed is thus about 1x1x2 microns. Indeed, all other demonstrations in the manuscript have minimum features above 2 microns. The paper contains no prints of common resolution test patterns, such as line arrays with varying widths and spacings.

Given that the commonly accepted criterion for resolution in 3D printing is the minimum layer thickness, I find the authors’ “sub-micron” printing claim unsubstantiated. At a minimum, the authors should clarify the 3D voxel size in the abstract. Preferably, they should demonstrate resolution test patterns with varying widths and spacings to quantify the resolution of their 3D process properly.

Lesser issues:

- The authors should clarify the meaning of “uncontrollable impurities” of structures printed from organosilicon resins (refs 20, 21, and 22). Given that the current manuscript lacks quantification of impurities, it seems misleading to hint that it improves on prior work in this respect.
- The claim that instruments required for two-photon lithography “incur costs in the tens of millions of dollars” is inaccurate. The list price for a common commercial two-photon instrument (Nanoscribe) is around USD 500,000.
- Fig. 1 c looks nice but adds no scientific value to the paper. The authors should at least clarify what sample it is and if it survived the treatment.
- The data presented in Fig 1 e is interesting, but it is hard to validate in its current form. It would be better to indicate directly in the graph what data comes from what paper (using the reference number as a data marker). The graph is only based on a handful of papers (one marker per paper (the best) is enough). As mentioned above, by the common definition of “Resolution” in the 3D printing world, this paper would not land on the north side of the 1-micron line. Also, even though printing time is an important aspect of a process, it ignores any required post-processing steps, which in the present case amount to 3800 minutes = 63 hours = 2.6 days (Fig S6b). This aspect should be mentioned in the text for a fairer comparison with slower printing methods that require minimal post-processing.
- The authors present many characterization results in the Supplementary Data but often do not comment on what conclusions they draw from the results. Also, information on the measured samples is often lacking. Each characterization result should be accompanied by a clear description of the measured

sample and the conclusions drawn from the results.

- The optical transmission spectra presented in Fig S10 c should include a suitable silica glass reference and ideally extend down to 300 nm (like Fig S1b) since the short wavelength behavior sets fused silica apart from other silica-based glasses.
- The number of significant digits reported in dimensions of “3.78 μm beam width” and “4.71 μm thickness” indicates an accuracy of ± 10 nm. However, no indication is given of how that is achieved.
- It should be made clear for all printed demonstrators if the structures are fully post-processed (sintered).
- Is the white light interferometer the right tool to quantify the surface roughness of the microlenses? With the limited x-y resolution of such an optical tool, is it possible to claim a surface roughness of 0.6 nm? Would not an AFM would be better suited to the task?
- The claim that this process allows the printing of “arbitrary structures” is inflated since the reported linear shrinkage of up to 52% severely limits what can be printed, as acknowledged by the authors in the design of the hydrophobic demonstrator: “the mushroom-inspired microstructures were regularly centered on the top of octet-truss microlattices rather than the solid substrate in order to be compatible with the shrinkage of the surface textures and substrates.” The claim of arbitrary printing capability should be moderated.
- I do not understand the sentence: “While the integration of fused silica glass and micro lattice architectures can significantly benefit from their ultrahigh stability, ultralight weight, and ultrahigh stiffness, these characteristics are not shared by both materials.” Please clarify.
- The discussion of the “weakest link” theory seems out of context and of little relevance.

Details:

- The English language needs copy-editing. In particular, the abstract.
- The illustration in Fig 1 a is too small to read. Also, the components should be labeled.
- The font on Fig 1 b and e is too tiny to read.
- The current paper does not define all acronyms used in Fig 1 e.
- The histogram in Fig S2 has non-uniform bins. This will yield a distorted view of the statistics.
- The transition to the paragraph starting with “Another most important step in obtaining high-quality silica is sintering.” is poor—the paragraph before discusses the “snake-fang” demonstrator, not any particular process step.
- What does the label “Original grayscale” mean in Fig 3 a? The green line is flat, i.e., no-grayscale.
- “Topological” should be “topographical” in Fig 3 caption.

Reviewer #2 (Remarks to the Author):

- What are the noteworthy results?

The authors demonstrate the 3D printing of silica glass 3D objects with size up to several millimetres and sub-micron resolution. Their results improve on the resolution achieved by Kotz et. al¹ where tens-of-microns resolution was demonstrated by SLA 3D printing of a fumed silica-based nanocomposite resin.

This result is achieved by developing a resin consisting of well-dispersed silica nanoparticles in an index-matched monomeric matrix. The proposed resin together with the O μ SL printing technology and optimized post-processing, yield crack-free millimetre-sized parts with sub-micron resolution, thereby bridging the gap between existing macroscale SLA and nanoscale TPL. The claim is justified because the authors demonstrate the printing of complex-shaped 3D part with sub micrometer features (fig 1d).

- Will the work be of significance to the field and related fields? How does it compare to the established literature? If the work is not original, please provide relevant references.

The work expands upon the current literature by achieving a sub-micrometer resolution using a printing technique (one-photon micro-stereolithography), whose resolution is comparable to lower-end TPP and whose printing speed is 6 orders of magnitude larger. Furthermore, they demonstrate, although not for the first time in the literature, potential applications in the fields of micro-optic (fig 3), micro-surfaces (fig 5) and microfluidic (Fig S11).

- Does the work support the conclusions and claims, or is additional evidence needed?

Yes.

- Are there any flaws in the data analysis, interpretation and conclusions? - Do these prohibit publication or require revision?

The authors carry out their resin characterization following the established methods and practices. However, some points should be addressed for the sake of reproducibility.

Particle diameter distribution: what solvent and concentrations were used for the analysis?

Rheology: What geometry was used for the rheological characterization?

Zeta potential: what solvent and concentrations were used for the analysis?

Compression test: what load cell and test conditions were used? Can the authors spend few words on the statistical relevance of the test performed?

- Is the methodology sound? Does the work meet the expected standards in your field?

See previous point.

- Is there enough detail provided in the methods for the work to be reproduced?

See previous point.

Additional Comments

The authors report using a commercial O μ SL system (BMF P μ SL nanoArch P130, China), for which the manufacturer provides a maximum resolution of 2 microns. As the authors claim achieving a 900nm resolution, they should specify that this is only achievable by exploiting the high shrinkage of their resins

(approximately 50% linear). Could the authors comment on this in relation to the possibility of producing large objects?

“Lastly, the desired femtosecond laser and the precise instruments still incur costs in the tens of millions of dollars”. I am not sure whether this statement is factually correct.

“one-photon micro-stereolithography (O μ SL) is another one-photon lithography technique that permits iterative polymerization in all corresponding regions upon UV light interaction with the photoresist” - can the authors spend few words describing how O μ SL differs from the DLP technology?

Bibliography

(1) Kotz, F.; Arnold, K.; Bauer, W.; Schild, D.; Keller, N.; Sachsenheimer, K.; Nargang, T. M.; Richter, C.; Helmer, D.; Rapp, B. E. Three-Dimensional Printing of Transparent Fused Silica Glass. *Nature* 2017, 544 (7650), 337–339. <https://doi.org/10.1038/nature22061>.

REVIEWER COMMENTS

Reviewer #1 (Remarks to the Author):

This manuscript reports the development of a precursor for additive manufacturing of three-dimensional silica glass objects by stereo-lithography. The main innovation is in the precursor formulation, which improves its optical and rheological properties, enabling the printing of smaller features than previously possible by stereolithography while maintaining benefits such as high printing speeds and large printing volumes. The authors demonstrate their method by printing three-dimensional glass objects of mm size with micrometric features. The new precursor formulation is of significant interest, and the demonstrations in the manuscript advance the state of the art. However, the central claim of this manuscript, and indeed its title, is the 3D printing of fused silica glass with sub-micron features.

Response: We feel great thanks for your professional review work on our article. As you are concerned, there are several problems that need to be addressed. According to your nice suggestions, we have made extensive corrections to our previous draft, the detailed corrections are listed below.

Major issues:

- Fused silica glass is pure silicon dioxide in amorphous (non-crystalline) form. Thus:

- o To prove that the 3D printed structures are made of silica, it is necessary to show that the bulk material of the printed structures only contains silicon and oxygen in the stoichiometric ratio 1:2 and to quantify any residual contaminants.

The authors employ two quantitative techniques (EDS and XPS) capable of determining stoichiometry and quantifying contaminants, but their presentation of the results is unclear.

1. In the case of EDS, the data is provided in Supplementary Data Fig. S9. First, there is no information on the sample. Information should be provided on what size of structure is being measured, if the measurement is done on the printed surface or in a cross-section, and in that case, how close to the printed surface the measurement is made. Furthermore, it should be made clear at what processing stage the sample was measured.

Response: Thanks for the reviewer's kind advice. According to the reviewer's comment, information regarding the printed and commercial fused silica glass is incorporated into page 13 in the text and is highlighted in red. A further EDS analysis is conducted on commercial fused silica glass and printed sample (both surface and cross-section), with corresponding results updated in Supplementary Data Fig. S12 to enhance comprehension. Therefore, the figure captions and citations are revised in accordance with the updated order.

“The measurement utilized a printed fused silica glass monolith with a diameter of 5.0

mm and a thickness of 0.5 mm. Additionally, a commercially available fused silica glass of identical dimensions was procured from Original Crystal Electronic Technology Co. Ltd (Shandong, China) as the control group. The SEM images in Supplementary Data Fig. S12b & c depict the printed fused silica glass exhibits an initial defect-free and uniform microstructure, both on the surface and in the cross-section, which is comparable to that of the commercial reference sample (Supplementary Data Fig. S12a). And the presented energy-dispersive spectroscopy (EDS) mappings further reveal the homogeneous distributions of Si and O elements across all three groups.

Supplementary Data Fig. S12. Electron microscopic and EDS mapping of the (a) commercial and O μ SL 3D printed fused silica glass ((b) surface & (c) cross-section); the elements are C, O and Si, respectively.”

2. Second, a representative EDS point spectrum should be included. In particular, the range from 0 to 2 kV is interesting to determine if any carbon contamination is present. The authors should also report the measured ratios of Si, O, and any detected contaminants, for example, carbon.

Response: The EDS spectra, ranging from 0 to 10 kV, are utilized for the measurement of Si, O and C element ratios in accordance with the comment. The corresponding results of commercial fused silica glass and printed fused silica glass (both surface and cross-section) are illustrated in Supplementary Data Fig. S13 with statement incorporated in page 13 in the text.

“A high level of consistency in binding energy, intensity, and atomic percentages of all elements were detected within the three groups, while the signals of carbon contaminants may arise from systematic noises or carbon deposition in the SEM chamber. The findings demonstrate that the final printed products exhibit an indistinguishable presence compared to commercial fused silica glass.

Supplementary Data Fig. S13. EDS spectra of the (a) commercial and O μ SL 3D printed fused silica glass ((b) surface & (c) cross-section).”

3. In the case of XPS, the data is provided in Supplementary Data Fig S10a. First, the sample information requested above should also be provided here. Second, a broad XPS spectrum (0 – 800 eV) should also be provided, and the measured ratios of Si, O, and any detected contaminants be reported. The authors should also clarify exactly how they conclude that XPS “identifies the final materials as stoichiometrically pure silica” based on their currently presented data. Finally, details should be provided on the “Commercial” reference sample.

Response: Thanks for the reviewer's kind advice. The information of the measured samples in the XPS corresponds to that mentioned above. The XPS spectrum ranging from 0 - 1350 eV was acquired for both printed and commercial fused silica glass, and the corresponding results are updated in page 14 and Supplementary Data Fig. S14a. The text is highlighted in red.

“The printed fused silica glass is composed of pure Si and O element, as confirmed by the full XPS analysis (0 - 1350 eV), which aligns with the characteristics observed in the commercially available reference sample. The fine spectroscopy results (0 - 35 eV for O1s, 97 - 107 eV for Si2p, and 526 - 536 eV for O2s) revealed that both the printed and commercial samples displayed comparable photoelectron binding energy and relative intensity, indicating their identical chemical state and structural composition.

Supplementary Data Fig. S14. Characterization of printed-sintered fused silica glass compared with its commercial counterpart: (a) full XPS spectrum and fine XPS spectra of each element ...”

Combining the XPS spectra with the XRD, Raman, TEM, and electron diffraction results (please find following response for details), it can be concluded that “identifies the final materials as stoichiometrically pure silica”. This conclusion also aligns with the findings reported by Kotz et al (Paragraph 4 Page 337, Paragraph 8 Extended Data Page 1, Fig. 2a and Extended Fig. S2a, *Nature*, 2017, 544 (7650), 337 - 339) and their previous studies.

4. To prove that the material is amorphous, one must show that it is without long-range order and doesn't contain crystallites. The authors employ XRD to argue that the sample is amorphous. The data is provided in Supplementary Data Fig S10b. First, the sample and reference information requested above is also missing for this

measurement and should be provided. Second, the measurement clearly shows a difference between the printed and reference samples. The reference sample has a prominent broad peak (also known as the amorphous halo) at about 22 degrees (2theta), with a slight hint at another broad peak around twice the angle (a half-order peak). The printed sample also has a prominent broad peak, but it is shifted to about 21 degrees, and the half-order peak is significantly more prominent than in the reference. A shift of the position of the amorphous halo to a lower angle in an X-ray diffraction (XRD) pattern of an amorphous material can indicate an increase in the average atomic spacing in the material and, thus, a less dense amorphous phase. Furthermore, the presence of a half-order peak can indicate a nanostructured material. Considering that the printed structures are made by sintering colloidal silica, the XRD data give reason to believe that the post-processing fails to achieve a homogenous fused silica glass. Indeed, the cross-sectional SEM of the sintered material in Supplementary Data Fig S8b seems to show a residual structure, although the contrast and resolution of that image are too poor to be certain. To convincingly resolve the question of the structure of the printed material, the XRD data should be complemented by transmission electron microscopy and electron diffraction data of a cross-section of a printed structure.

Response: Thanks for the reviewer’s careful observation. The XRD tests were repeated on both commercial and printed fused silica glass. Straight line subtraction and deconvolution were also employed to facilitate comparison of the two results. The data is updated in Supplementary Data Fig. S14b.

“**Supplementary Data Fig. S14.** Characterization of printed-sintered fused silica glass compared with its commercial counterpart: ... (b) XRD spectrum ...”

Based on the comment, we further confirm the amorphous phase of the printed fused silica glass through TEM and electron diffraction. The result is incorporated into page

15 in the text and is highlighted in red. Corresponding images are also updated in Supplementary Data Fig. S16 to enhance comprehension.

“The transmission electron microscopy (TEM) images in Supplementary Data Fig. S16a & b demonstrate the dense nature of the printed fused silica glass structures, exhibiting an absence of discernible pores or cracks. The Supplementary Data Fig. S16c-e presents energy-dispersive spectroscopy mapping, which reveals the homogeneous distributions of Si and O elements. The diffraction patterns (inset of Supplementary Data Fig. S16b) corresponding to the amorphous phase exhibit excellent agreement with the X-ray diffraction analysis.

Supplementary Data Fig. S16. **a** The printed fused silica glass after Focus Ion Beam (FIB) milling. **b** TEM image and electron diffraction pattern of the printed fused silica glass. EDS mappings of (c) total, (d) Si, (e) O, and (f) Pt element.”

5. The claims of patterning “sub-micron features,” as stated in the title of the manuscript, and sub-micron “resolution,” as indicated in Fig. 1 e, are based on the structure shown in Fig. 1d (lower left). The figure shows an isolated line with about 900 nm width in one in-plane dimension. The thickness of the features printed with the method (z-axis resolution) is, however, in all cases above 2 microns. The minimum isolated voxel that can be printed is thus about 1x1x2 microns.

Response: We appreciate the reviewer's thorough evaluation and analysis. The cured thickness in the z-axis is definitely one of the most important parameters for assessing the resolution of a printing system. This parameter is primarily determined by the photosensitive properties of the precursor resin and the precision in positioning of the z-axis motor. The well-prepared precursor exhibits desirable absorption coefficients, effectively preventing shape errors caused by overexposure and achieving a uniform printing thickness of approximately 2 μm (Supplementary Data Fig. S5) in the growth

direction of the sample (z-axis). The additional volumetric shrinkage facilitates a further reduction in the achievable minimum feature size of the printed layer, resulting a thickness below 1 μm (the zoom-in SEM image of the snake-fang bio-inspired microneedle, Fig. 2d) after the final sintering process. The SEM images of monolayer thickness for the as-printed, before-sintered, and after-sintered samples have also been included in Supplementary Data Fig. S6.

Supplementary Data Fig. S6. The Zoom-in SEM images of the snake-fang bio-inspired microneedle: (a) as-printed, (b) before-sintered, and (c) after-sintered.

6. Indeed, all other demonstrations in the manuscript have minimum features above 2 microns. The paper contains no prints of common resolution test patterns, such as line arrays with varying widths and spacings.

Response: Thanks for the reviewer's kind suggestion. To further validate the feasibility of sub-micron manufacturing, we have designed an additional resolution test pattern consisting of a 15×10 column array with line arrays affixed to their tops. The corresponding SEM images are presented in Supplementary Data Fig. S10, while the statement is incorporated on page 8 of the text and highlighted in red.

“The resolution was further confirmed by constructing an additional 15×10 column array with line arrays affixed on top (Supplementary Data Fig. S10a-c). The line array comprises four lines in both vertical and horizontal orientations, respectively arranged with varying horizontal spacing and vertical widths. The proposed technology enables simultaneous sub-micron resolution in both width and distance (Supplementary Data Fig. S10d), with features exhibiting a line spacing ranging from approximately 800 nm to 4 μm as well as a line width ranging from approximately 800 nm to 3 μm .

Supplementary Data Fig. S10. (a) Optical & (b-d) electron microscopic images of a 15×10 column array affixed line arrays on the tops, suggesting the sub-micron features in both line width and spacing.

Given that the commonly accepted criterion for resolution in 3D printing is the minimum layer thickness, I find the authors' "sub-micron" printing claim unsubstantiated. At a minimum, the authors should clarify the 3D voxel size in the abstract. Preferably, they should demonstrate resolution test patterns with varying widths and spacings to quantify the resolution of their 3D process properly.

Response: We appreciate the reviewer's thorough evaluation and analysis. The sub-micron printing of fused silica glass in both in-plane and growing directions, as well as other resolution test patterns with widths and spacing down to 800 nm, have been further validated. Please refer to Response 5 & 6 for detailed information.

Lesser issues:

7. The authors should clarify the meaning of "uncontrollable impurities" of structures printed from organosilicon resins (refs 20, 21, and 22). Given that the current manuscript lacks quantification of impurities, it seems misleading to hint that it improves on prior work in this respect.

Response: Thanks for the reviewer's thorough evaluation. The presence of impurities in the manuscripts is supported by the available data. A discernible peak corresponding to the carbon-carbon bond (1630 cm^{-1}) exists in the Raman spectra (Paragraph 4 Page 5 of 7 and Fig. 2D, Bauer et al., *Science*, 2023, 380, 960-966), regardless of the sintering temperature employed (from $650 \text{ }^\circ\text{C}$ to $1200 \text{ }^\circ\text{C}$). It can be attributed to carbon contaminants originating from the organos resins during the thermal treatment, whereas

no corresponding peak was observed in their commercial reference sample (Fig. 2D, Bauer et al., *Science*, 2023, 380, 960-966). Moreover, the uncontrollable impurities also lead to compromised optical properties, leading to a transmittance deterioration approximately 10% compared to the reference glass (Fig. 4b, Moore et al., *Nature Materials*, 2020, 19, 212 - 217). The compositional inhomogeneity led to interfacial scattering, resulting in a whitish and opaque appearance of the sample (Fig. 3I, Fig. 3J, and Fig. 4a, Moore et al., *Nature Materials*, 2020, 19, 212 - 217).

Therefore, we conducted additional measurements of Raman spectra for both printed fused silica glass and the commercial reference sample in order to validate our advancements in this particular aspect. The corresponding Raman images are presented in Supplementary Data Fig. S16c, while the sentences are incorporated on page 14 - 15 of the text and highlighted in red.

“The Raman spectroscopy measurement were performed to characterize the material's structural properties (Supplementary Data Fig. S16c). In the context, the ω_1 bands (cm^{-1}) and ω_3 bands (cm^{-1}) relate to bending vibration of the $\text{Si}(\text{O}_{1/2})_4$ tetrahedrons' Si-O-Si bridges, and ω_4 bands (cm^{-1}) can be ascribed to the stretching motion of the Si-O bonds. The D_1 bands (cm^{-1}) and D_2 bands (cm^{-1}) correspond to the symmetric stretching of silicon-oxygen ring molecules. The spectrum exhibits no additional peaks, confirming the absence of any residual impurities in the printed fused silica glass. The excellent consistency observed in the spectrum of the printed fused silica glass, when compared to that of the commercial reference sample, confirms its composition as pure silicon dioxide, thus aligning with the composition the commercial fused silica glass.

Supplementary Data Fig. S16. Characterization of printed-sintered fused silica glass compared with its commercial counterpart: ... (c) Raman spectra.”

8. The claim that instruments required for two-photon lithography “incur costs in the

tens of millions of dollars” is inaccurate. The list price for a common commercial two-photon instrument (Nanoscribe) is around USD 500,000.

Response: Thanks for the reviewer’s kind advice and sorry for our obscure writing. The claim here aims to convey that the instruments necessitate tens of millions of HKD (Hong Kong dollars). While the cost of high-resolution two-photon instruments (Quantun X, Nanoscribe, Germany) exceeds one million USD. Now the statement has been revised as a new sentence in the text and highlighted in red: “... Lastly, the desired femtosecond laser and the precise instruments still cost considerably higher dollars.”

9. Fig. 1 c looks nice but adds no scientific value to the paper. The authors should at least clarify what sample it is and if it survived the treatment.

Response: Thanks for the reviewer’s kind suggestion. The suggestion has been implemented by incorporating a paragraph into page 8 to enhance comprehension, which is also highlighted in red. “Moreover, the O μ SL 3D printed fused silica glass exhibits exceptional transparency even when subjected to thermal shocks of several hundred degrees, as shown in Fig. 1c. The printed fused silica glass micro-lens array (MLA) sample remained intact following instant heating and subsequent cooling, while the organic glasses are unable to withstand such high temperature, resulting in either softening or combustion.”

10. The data presented in Fig 1 e is interesting, but it is hard to validate in its current form. It would be better to indicate directly in the graph what data comes from what paper (using the reference number as a data marker). The graph is only based on a handful of papers (one marker per paper (the best) is enough). As mentioned above, by the common definition of “Resolution” in the 3D printing world, this paper would not land on the north side of the 1-micron line.

Response: Thanks for the reviewer’s kind advice. The data is classified according to the reviewer’s comment, with the reference number as the marker. The figure now encompasses a greater number of papers to facilitate a more comprehensive comparison. In views that this work definitely achieves sub-micron resolutions (Fig. 1d, Fig. 2d, and Response 5 & 6), the marks do not need further revise and should remain unchanged on their original site. The corresponding data is updated in Fig. 1e.

“

Fig. 1: 3D printing transparent fused silica glass with one-photon lithography. ...
e Plot of the finest resolution against the maximum printing speed of the 3D printed transparent fused silica glass in our work, together with the data of other reported 3D-printed transparent fused silica glass for comparison.”

11. Also, even though printing time is an important aspect of a process, it ignores any required post-processing steps, which in the present case amount to 3800 minutes = 63 hours = 2.6 days (Fig S6b). This aspect should be mentioned in the text for a fairer comparison with slower printing methods that require minimal post-processing.

Response: Thanks for the reviewer’s suggestion. The consideration of time costs associated with post-processing is important in the discuss on additive manufacturing. However, employing diverse post-processing techniques results in varying time costs, and emerging sintering methods such as Spark Plasma Sintering (SPS) and Ultrafast High-temperature sintering (UHS) dramatically reduce the duration. On the other hand, diverse printing processes give rise to varying resolutions, forming dimensions, and printing speeds, thus directly influencing the ultimate manufacturing time. Therefore, we maintain that the printing process holds utmost significance, and the discuss should revolve around the printing time.

12. The authors present many characterization results in the Supplementary Data but often do not comment on what conclusions they draw from the results. Also, information on the measured samples is often lacking. Each characterization result should be accompanied by a clear description of the measured sample and the conclusions drawn from the results.

Response: We thought that the characterizations in Supplementary Data (particle size measurement, viscosity measurement, zeta potential measurement, TGA measurement, etc.) is recognized as standard operating procedures, so the detailed operating procedures are not included in the text. However, according to the reviewer’s comment, information regarding the measured samples and operation processes are added in chapter “Characterization”, page 28 - 29 in the text and highlighted in red. “Deionized water was employed as the solvent for particle size distribution analysis, while ultra-pure water was utilized as the solvent for Zeta potential measurements.” “The rheological properties were evaluated by measuring the dynamic viscosities as the functions of shear rate and temperature, using the fixtures with a diameter of 60 mm and geometries of cone/plate and parallel plates, respectively.” “The microlattices were subjected to uniaxial compression at room temperature with a specified strain rate of 10^{-3} s^{-1} , using the testing system equipped with a load cell of 100 N.”

13. The optical transmission spectra presented in Fig S10 c should include a suitable silica glass reference and ideally extend down to 300 nm (like Fig S1b) since the short wavelength behavior sets fused silica apart from other silica-based glasses.

Response: Thanks for the reviewer’s kind advice. The uv-vis transmission spectra have now updated with data extend down to 300 nm. The corresponding data is updated in

Supplementary Data Fig. S15.

Supplementary Data Fig. S15. UV-Vis transmission spectra of a printed fused silica glass monolith with diameter of 5 mm and a thickness of 0.5 mm.”

14. The number of significant digits reported in dimensions of “3.78 μm beam width” and “4.71 μm thickness” indicates an accuracy of +/- 10 nm. However, no indication is given of how that is achieved.

Response: The two sizes are measured during SEM characterizations (see the figure below), and both the accuracy and number of significant digits are directly determined by the image measurement software.

15. It should be made clear for all printed demonstrators if the structures are fully post-processed (sintered).

Response: A sentence is added in page 11 to state that all the sample are subjected the full post-processing procedure. “All these O μ SL 3D-printed structures have undergone complete post-processing and final sintering treatment.”

16. Is the white light interferometer the right tool to quantify the surface roughness of the microlenses? With the limited x-y resolution of such an optical tool, is it possible to claim a surface roughness of 0.6 nm? Would not an AFM would be better suited to the task.

Response: The atomic force microscopy (AFM) is extensively employed for the

characterization of surface morphology of planar samples. However, the presence surface curvature in our lens sample poses a challenge for measurement using AFM, as it may potentially cause damage to both the sample and the AFM probe. On the other hand, the white light interference is one of the most prevalent techniques in the optical measurement due to its non-contact nature and non-destructive impact on the sample surface. Moreover, the white light interference also exhibits the characteristics of low noise and high accuracy, with vertical direction precision of up to 0.1nm. In conclusion, white light interference proves to be a more appropriate testing method in this task compared to AFM.

17. The claim that this process allows the printing of “arbitrary structures” is inflated since the reported linear shrinkage of up to 52% severely limits what can be printed, as acknowledged by the authors in the design of the hydrophobic demonstrator: “the mushroom-inspired microstructures were regularly centered on the top of octet-truss microlattices rather than the solid substrate in order to be compatible with the shrinkage of the surface textures and substrates.” The claim of arbitrary printing capability should be moderated.

Response: Thanks for the reviewer’s advice. Firstly, A large number of studies have been reported on the fabrication of superhydrophobic microstructures on solid substrates. Our proposed O μ SL 3D printing technologies also has the ability to manufacture such components. Please refer to the drawings below.

Second, the study of lattice structures in metamaterials has been widely explored. However, the current focus in hydrophilic modification of lattice structures primarily revolves around material modifications. Here we explored a novel approach in combining the bio-inspired microstructures with microlattice architectures, aiming to demonstrate the feasibility of modifying the hydrophilicity of lattice metamaterials from the structural perspective. Therefore, the claim on the arbitrary printing capability of O μ SL 3D printing technologies does not require any additional moderation.

18. I do not understand the sentence: “While the integration of fused silica glass and micro lattice architectures can significantly benefit from their ultrahigh stability, ultralight weight, and ultrahigh stiffness, these characteristics are not shared by

both materials.” Please clarify.

Response: The sentence here is to state that the integration of fused silica glass and micro lattice architectures endows metamaterials with exceptional comprehensive properties (excellent transparency, ultrahigh stability, ultralight weight, and ultrahigh stiffness), and those exceptional comprehensive properties is unattainable through any individual constituent.

19. The discussion of the “weakest link” theory seems out of context and of little relevance.

Response: Weibull's “weakest link” theory has been extensively employed to model the strength distribution and the failure probability in a serial system comprising multiple identical brittle solids. From an architectural point of view, the minimization of feature size achieved by the proposed O μ SL 3D printing technologies have brought unexpected benefits to the mechanical properties of fused silica glass, including increased strength over its bulk counterpart. This phenomenon is typically known as the ‘smaller is stronger’ tendency and can be attributed to fewer defects, stronger surface interactions, and various other mechanisms. Combined with the Weibull’s “weakest link” theory, O μ SL 3D printing technology enhances structural precision to a greater extent, leading to exponential improvements in strut performance (Equation (3)). Consequently, this significantly diminishes the probability of system failure (Equation (2)) and contributes substantially to the advancement of robust fused silica glass mechanical metamaterials.

Details:

20. The English language needs copy-editing. In particular, the abstract.

Response: Thanks to the reviewer’s kind advice. Now the language is further polished.

21. The illustration in Fig 1 a is too small to read. Also, the components should be labeled.

Response: The image has been updated with a higher resolution. An additional schematic with all components labeled is added in Supplementary Data Fig. S8.to facilitate comprehension.

“**Supplementary Data Fig. S8.** The optical setup sketch O μ SL 3D printing process of miniature Hong Kong dioramas.”

22. The font on Fig 1 b and e is too tiny to read.

Response: Larger font sizes are employed. The corresponding details were updated in Fig. 1b and 1e.

23. The current paper does not define all acronyms used in Fig 1 e.

Response: Thanks to the reviewer’s careful observation. All acronyms are now added in the text.

24. The histogram in Fig S2 has non-uniform bins. This will yield a distorted view of the statistics.

Response: Thanks to the reviewer’s kind advice. Uniform bins are now employed in Supplementary Data Fig. S2.

“**Supplementary Data Fig. S2.** Particle diameter distribution of functionalized colloidal silica nanoparticles in polymerizable monomeric matrix.”

25. The transition to the paragraph starting with “Another most important step in obtaining high-quality silica is sintering.” is poor—the paragraph before discusses the “snake-fang” demonstrator, not any particular process step.

Response: Thanks to the reviewer's kind advice. The sentence is now revised for the better transition in page 12 in the text and highlighted in red. "Following the high-precision O μ SL of arbitrary 3D micro-/nano- architectures, sintering is another most important step in obtaining high-quality fused silica glass components."

26. What does the label "Original grayscale" mean in Fig 3 a? The green line is flat, i.e., no-grayscale.

Response: Thanks for the reviewer's careful observation. Before O μ SL 3D printing process, the 3D model would be firstly transformed into a succession of 2D digital profiles. The 2D profiles consist of individual pixels, each represented by a grayscale value of either 0 (pure black) or 255 (pure white). The label "Original grayscale" in Fig. 3a denotes the profile without modulating its grayscale values.

27. "Topological" should be "topographical" in Fig 3 caption.

Response: Thanks to the reviewer's careful observation. The word has been rectified.

Reviewer #2 (Remarks to the Author):

- What are the noteworthy results?

The authors demonstrate the 3D printing of silica glass 3D objects with size up to several millimetres and sub-micron resolution. Their results improve on the resolution achieved by Kotz et. al where tens-of-microns resolution was demonstrated by SLA 3D printing of a fumed silica-based nanocomposite resin. This result is achieved by developing a resin consisting of well-dispersed silica nanoparticles in an index-matched monomeric matrix. The proposed resin together with the O μ SL printing technology and optimized post-processing, yield crack-free millimeter-sized parts with sub-micron resolution, thereby bridging the gap between existing macroscale SLA and nanoscale TPL. The claim is justified because the authors demonstrate the printing of complex-shaped 3D part with sub micrometer features (fig 1d).

- Will the work be of significance to the field and related fields? How does it compare to the established literature? If the work is not original, please provide relevant references.

The work expands upon the current literature by achieving a sub-micrometer resolution using a printing technique (one-photon micro-stereolithography), whose resolution is comparable to lower-end TPP and whose printing speed is 6 orders of magnitude larger. Furthermore, they demonstrate, although not for the first time in the literature, potential applications in the fields of micro-optic (fig 3), micro-surfaces (fig 5) and microfluidic (Fig S11).

Response: We greatly appreciate your professional review of our article. We have taken into consideration your valuable suggestions and made corresponding revisions to our previous draft. The detailed corrections are listed.

- Does the work support the conclusions and claims, or is additional evidence needed?

Yes.

- Are there any flaws in the data analysis, interpretation and conclusions? - Do these prohibit publication or require revision?

28. The authors carry out their resin characterization following the established methods and practices. However, some points should be addresses for the sake of reproducibility.

Particle diameter distribution: what solvent and concentrations were used for the analysis?

Rheology: What geometry was used for the rheological characterization?

Zeta potential: what solvent and concentrations were used for the analysis?

Compression test: what load cell and test conditions were used? Can the authors spend few words on the statistical relevance of the test preformed?

Response: Thanks for the reviewer's kind advice. We thought that the characterizations in Supplementary Data (particle size measurement, viscosity measurement, zeta potential measurement, TGA measurement, etc.) is recognized as standard operating procedures, so the detailed operating procedures are not included in the text. However, according to the reviewer's comment, information regarding the measured samples and operation processes are added in chapter "Characterization", page 28 - 29 in the text and highlighted in red. "Deionized water was employed as the solvent for particle size distribution analysis, while ultra-pure water was utilized as the solvent for Zeta potential measurements." "The rheological properties were evaluated by measuring the dynamic viscosities as the functions of shear rate and temperature, using the fixtures with a diameter of 60 mm and geometries of cone/plate and parallel plates, respectively." "The microlattices were subjected to uniaxial compression at room temperature with a specified strain rate of 10^{-3} s^{-1} , using the testing system equipped with a load cell of 100 N." The sets of data in the mechanical properties test correspond to the same printed lattice structure, albeit exhibiting slight variations attributed to manufacturing process. Hence, there exists an absence of an additional variable to facilitate the discussion on statistical relevance in mechanical performance. However, we have now included a detailed set of parameters for the tested O μ SL 3D printed fused silica glass microlattice (density, strut diameters, and maximum compressive strength) in Supplementary Table S1, hoping that can enhance the reader's comprehension.

- Is the methodology sound? Does the work meet the expected standards in your field?

See previous point.

- Is there enough detail provided in the methods for the work to be reproduced?

See previous point.

Additional Comments

29. The authors report using a commercial O μ SL system (BMF P μ SL nanoArch P130, China), for which the manufacturer provides a maximum resolution of 2 microns. As the authors claim achieving a 900nm resolution, they should specify that this is only achievable by exploiting the high shrinkage of their resins (approximately 50% linear). Could the authors comment on this in relation to the possibility of producing large objects?

Response: In our experiments, the 900nm resolution is produced by high shrinkage of the resins cured using a 2 μ m-capable instrument. However, the potential resolution by one-photon curing can exceed this value. As shown in the figure below, our resin can produce sub-micron features when cured using a photolithography instrument with intrinsic sub-micron resolution.

For the other question, the proposed O μ SL 3D printing technique enables an increase in resolution while maintaining the capability to print larger or multiple parts with high tolerance. The motors in X, Y directions can be incorporated with the platform, enabling the creation of multiple printing areas for part or multiple parts, depending on their size. This capability allows for the production of either a larger integrated component by the projection zones, or multiple smaller components through efficient nesting.

30. “Lastly, the desired femtosecond laser and the precise instruments still incur costs in the tens of millions of dollars”. I am not sure whether this statement is factually correct.

Response: Thanks for the reviewer’s kind advice and sorry for our obscure writing. The claim here aims to convey that the instruments necessitate tens of millions of HKD (Hong Kong dollars). While the cost of high-resolution two-photon instruments (Quantun X, Nanoscribe, Germany) exceeds one million USD. Now the statement has been revised as a new sentence in the text and highlighted in red: “... Lastly, the desired femtosecond laser and the precise instruments still cost considerably higher dollars.”

31. “one-photon micro-stereolithography (O μ SL) is another one-photon lithography technique that permits iterative polymerization in all corresponding regions upon UV light interaction with the photoresist” - can the authors spend few words

describing how O μ SL differs from the DLP technology?

Bibliography

- (1) Kotz, F.; Arnold, K.; Bauer, W.; Schild, D.; Keller, N.; Sachsenheimer, K.; Nargang, T. M.; Richter, C.; Helmer, D.; Rapp, B. E. Three-Dimensional Printing of Transparent Fused Silica Glass. *Nature* 2017, 544 (7650), 337–339. <https://doi.org/10.1038/nature22061>.

Response: Thanks to the reviewer’s kind advice. A paragraph is added to page 2 - 3 to facilitate comprehension. “By positioning a high-precision reduction lens between the projector and the resin tank, the resolution is finely modulated to the desired levels. In addition, O μ SL 3D printing is carried out in top-down direction, which reduces the need for support structures and protects intricate details from damage. Therefore, the O μ SL systems can achieve a printing resolution as low as 2 μ m and exhibit a dimensional tolerance of up to \pm 2 μ m.”

Reviewers' Comments:

Reviewer #1 (Remarks to the Author):

The authors have provided additional details regarding material characterization and printing resolution. However, the authors have not sufficiently addressed my criticism of their overclaiming of the printing resolution of their method. In fact, the additional data supplied only strengthens my opinion. To state my position succinctly: The current title of this manuscript is misleading. A more representative title is: "One-photon Three-Dimensional Printing of Fused Silica Glass with Micrometric Resolution".

Details follow below.

Material characteristics:

EDS analysis:

- The choice of color for carbon in the representation of the EDS spectra in Figure S12 is poor. The dark brown colour has poor contrast to the black background. Please use a colour with better contrast (e.g., magenta).
- Information is still missing on how close to the surface the cross-sectional data was collected.
- The information on the commercial fused silica reference sample is still insufficient. Only the name of the supplier is stated, but no details on the type of glass. A part number and a manufacturer-specified purity are necessary to understand the relevance of this reference sample.
- The new EDS point spectra reported in Supplementary Data Fig. S13 are puzzling. All three samples (including the commercial fused silica reference) display 9.7% carbon content. The authors blame this on "systematic noises or carbon deposition in the SEM chamber."

First, the term "systematic noises" has no scientific meaning and certainly doesn't count as an explanation for this result. The second explanation sounds more plausible, but this level of carbon contamination is extreme. Were the samples carbon coated before imaging (as sometimes used to avoid charging)? If so, this should be clearly stated in the paper.

Given this high level of background carbon contamination, nothing can be learned from this measurement about carbon contamination in the printed samples. The authors should redo this measurement, employing a suitable cleaning procedure (e.g., Piranha etch followed by dilute HF dip). Also, avoid focusing and parking the beam at the same spot for long, since this will accumulate carbon on the surface ("burn a contamination spot").

XPS:

- The broad spectrum in Fig. S14 a contains many unidentified peaks. The authors must identify and label the peaks visible in the spectrum.
- The authors must clarify how the energy axis was calibrated in the measurement, as this is a challenge for XPS of electrically insulating samples: see [10.1002/anie.201916000](https://doi.org/10.1002/anie.201916000)
- The XPS data curves of the printed and the reference samples should be offset in the y-axis direction to facilitate a comparison of the peak heights (as plotted now, the data of the printed sample hides the data of the reference sample).
- The authors still do not report stoichiometric ratios obtained from the XPS measurement. The authors must provide this data for the printed and the reference samples.
- The authors claim that the Si2p, O1s, and O2s peaks of the printed and reference sample "displayed

comparable photoelectron binding energy and relative intensity.” This is, however, impossible for the reader to verify since they have normalized each peak separately. The same normalization must be applied to each sample’s peaks to facilitate comparison of relative peak heights.

- There seems to be an energy offset between O1s and O2s peaks of the printed and reference samples of about 0.1 eVs and 0.5 eVs, respectively. The authors must clarify if this is a significant difference.

XRD:

- The authors have not responded to the subject of my criticism of the XRD data presented in Supplementary Data Fig S10b in the original submission. In the revised submission, the authors simply present new XRD data in Supplementary Data Fig. S14b, commenting that the data has been subjected to “straight line subtraction and deconvolution.” These terms have no physical meaning in the analysis of XRD data. The authors must clarify what was done differently in the new measurement and analysis and explain why their result now differs from what they presented in the original submission.

TEM:

- The authors have performed TEM and EDS measurements of a cross-section of a printed sample. However, the authors only show the Si, O, and Pt distribution without any quantification. Also, the primary possible contaminant, carbon (C), is left out of the analysis. The authors should provide the quantification of the Si/O stoichiometry and the amount of carbon measured.

Printing resolution:

- The authors write in their abstract:

“our study demonstrates ... that 3D printing based on one-photon micro-stereolithography (O μ SL) permits the flexible creation of transparent and high performance fused silica glass components with complex, 3D SUB-MICRON ARCHITECTURES”. (emphasis added)

and

“The methodology facilitates the construction of fused silica glass components with ARBITRARY 3D GEOMETRIES FEATURING FINE DETAILS AS SMALL AS 0.8 μ M”. (emphasis added)

In my opinion, the authors have not demonstrated this in their work. Furthermore, they ignored my explicit request to clearly state in the abstract the dimensions of the smallest voxel that can be printed. From supplementary Figure S5 b, we see that the printed layer thickness is about 2.5 μ m. Judging from the new supplementary Figure S6, the layers shrink to 1.9 μ m upon decarbonization at 600 °C. The layers then reach an ultimate average thickness of 0.9 μ m after sintering at 1050 °C, but the layer-to-layer variation in thickness is a few hundred of nanometres. Furthermore, after the high-temperature sintering, a significant rounding of the structures is visible, further limiting the resolution. This rounding effect is also visible in the new Figure S10.

Looking more closely at Figure S10, it is clear that the rounding after sintering is a severe limitation. It is thus unclear why the authors write in the caption of Figure S10 that it is “suggesting the sub-micron features in both line width and spacing,” I don’t observe anything resolved in the image with dimensions below one micron. In Figure S10 d, the smallest resolved line is 1.4 μ m, and the smallest resolved gap is 1.2 μ m. Furthermore, the authors failed to state what line and gap design they aimed to print—finally, the patterns in Figs. S6 and S10 don’t contain any suspended lines. Without that, one cannot state anything about the resolution of a 3D printing process that allows the creation of “arbitrary 3d geometries,” in the author’s own words.

From my viewpoint, the authors cherry-pick the smallest obtained values in one dimension without stating the values for the two others at the same conditions. Judging from the limited information

provided, I estimate that this method can print a voxel of size about $2 \times 2 \times 3 \mu\text{m}^3$ before post-processing, which shrinks to about $1 \times 1 \times 1 \mu\text{m}^3$ after post-processing. Any claims of sub-micron patterning capability remain unfounded.

- The placement of the white scale bar on the bright (charging) area in Fig. S10 d makes it very difficult to see. Use a black scale bar or move the white scale bar to a dark background.
- The authors must state CLEARLY in their abstract the three dimensions of the smallest resolved voxel BEFORE and AFTER post-processing. The obligatory high-temperature sintering is a severe limitation for printing on a substrate. Only very few substrates survive heating to $1050 \text{ }^\circ\text{C}$, and the shrinkage of the printed structure with respect to the substrate will cause deformation and even cracking and detachment. Without this information clearly stated in the abstract, it is impossible to compare the advance of this work to previous similar work, some of which require no post-processing.
- Since the printing approach applied here is based on linear absorption of light at a wavelength of 405 nm , it would be highly unexpected to achieve sub-micron resolution with this technique, and indeed, the results presented do not support such a claim. The only 3D printing methods convincingly demonstrating sub-micron resolution are based on non-linear (multiphoton) absorption.

Reviewer #2 (Remarks to the Author):

the revision is thorough and well executed, compliments to the authors, im happy to recommend for publication

Response to reviewers' comments:

Reviewer #1 (Remarks to the Author):

The authors have provided additional details regarding material characterization and printing resolution. However, the authors have not sufficiently addressed my criticism of their overclaiming of the printing resolution of their method. In fact, the additional data supplied only strengthens my opinion. To state my position succinctly: The current title of this manuscript is misleading. A more representative title is:

“One-photon Three-Dimensional Printing of Fused Silica Glass with Micrometric Resolution”.

Response: We would like to extend our sincere appreciation for your expert review of our article. According to your and the editor's comments, the scientific value of our manuscript is constrained by two major shortcomings: printing resolution and material characterization. Our apologies for the ambiguity that has been introduced into our work description and writing. We have added additional evidence to support our conclusions in this version, which includes *evidence of intrinsic sub-micron features*, and *the addition and explanation of characterization*. It is anticipated that the findings will contribute to the manuscript's scientific framework. In response to the recent comments, we have revised the manuscript accordingly, and the corresponding responses are presented in a sequential fashion below.

Details follow below.

Material characteristics:

EDS analysis:

1. The choice of color for carbon in the representation of the EDS spectra in Figure S12 is poor. The dark brown colour has poor contrast to the black background. Please use a colour with better contrast (e.g., magenta).

Response: Thanks for the reviewer's kind suggestions. The positive red color was selected in our EDS tests to represent the distribution of carbon elements. Please refer

to the original EDS figure below.

The decrease in contrast can be ascribed to the compression of the image file format. Your kind reminder is greatly appreciated. We now further increase the brightness for a better contrast, with corresponding results updated in Supplementary Data Fig. S13.

“

Supplementary Data Fig. S13. Electron microscopic and EDS mapping of the (a) commercial and OμSL 3D printed fused silica glass ((b) surface & (c) cross-section); the elements are C, O and Si, respectively.”

2. Information is still missing on how close to the surface the cross-sectional data was

collected.

Response: Thanks for the reviewer's kind suggestion and we apologize for our unclear writing. The suggestion has been implemented by incorporating a paragraph into Page 14 to enhance comprehension, which is also highlighted in red. "... The cross-sectional measurement of the printed fused silica glass was conducted at a location 250 μm beneath the surface, from the center plane of the 0.5-mm-thick monolith."

3. The information on the commercial fused silica reference sample is still insufficient. Only the name of the supplier is stated, but no details on the type of glass. A part number and a manufacturer-specified purity are necessary to understand the relevance of this reference sample.

Response: We express our gratitude for your thoughtful guidance and apologize for the obscurity of our writing. More details of the commercial fused silica glass reference sample are now measured and included in Page 14, which is also highlighted in red. "... Additionally, the commercially available fused silica glass monoliths (JGS-1 Grade, very high content of pure silica ($\text{SiO}_2 \geq 99.9999\%$), DSP Surface Finished, TTV < 20 μm , Bow/Warp < 60 μm , Top Side Ra < 1 nm) of identical dimensions were procured from Original Crystal Electronic Technology Co. Ltd (Shandong, China) as the control group."

• The new EDS point spectra reported in Supplementary Data Fig. S13 are puzzling. All three samples (including the commercial fused silica reference) display 9.7% carbon content. The authors blame this on "systematic noises or carbon deposition in the SEM chamber."

4. First, the term "systematic noises" has no scientific meaning and certainly doesn't count as an explanation for this result. The second explanation sounds more plausible, but this level of carbon contamination is extreme. Were the samples carbon coated before imaging (as sometimes used to avoid charging)? If so, this should be clearly stated in the paper.

Response: We appreciate your comprehensive assessment and analysis, and extend our

apologies for any confusion that may have been introduced by our composition. We mistakenly used the term “systematic noises” to describe the inherent limitations of EDS in accurately quantifying light elements like carbon. Quantitative EDS analysis of light elements is difficult because of the severe self-absorption of low energy X-rays, poor excitation efficiency, and low fluorescence yield^{1, 2}.

Thank you for the question about carbon coating. The samples were not subjected to carbon coating before the SEM experiments. And there are additional factors that may affect the accuracy of EDS measurements in SEM, especially for light elements. We have analyzed and identified three contributing factors: i) Before SEM, the specimens were sputtering coated with platinum (Pt) to obtain a finer microstructure^{3, 4}. The coating target consists of platinum complexes⁵, which are organometallic composition. Therefore, insufficient decomposition of organometallic composition during platinum deposition results in the introduction of carbon contaminants. ii) We used conductive carbon tape to position the specimens on the stage during the SEM-EDS measurements. For EDS analysis, the electron beam’s effect on the sample surface and its penetration into the depth produces X-rays that are characteristic of the elements². Therefore, using the conductive carbon tape as the substrate would also produce the carbon signal. iii) Conducting SEM-EDS tests in an absolute vacuum environment is impossible^{2, 6}. The presence of residual carbon dioxide (CO₂) within the chamber also results in the generation of the carbon signal.

We have now removed the inaccurate terminologies and sentences from the text to prevent confusion and misunderstanding. Moreover, a new sentence is added into Page 15 with corresponding reference updated, which is also highlighted in red. “... Despite the well-known inherent limitations of EDS in qualifying light elements^{36, 37, 38, 39, 40, 41}, the findings demonstrate that the quality and purity of the final printed products are well comparable to those of commercially high-quality fused silica glass.”

5. Given this high level of background carbon contamination, nothing can be learned from this measurement about carbon contamination in the printed samples. The authors should redo this measurement, employing a suitable cleaning procedure

(e.g., Piranha etch followed by dilute HF dip). Also, avoid focusing and parking the beam at the same spot for long, since this will accumulate carbon on the surface (“burn a contamination spot”).

Response: We appreciate your insightful concerns. Our manuscript presents a new technique for fabricating fused silica glass 3D-architectures with sub-micron features, and thus we aim to characterize the most primitive materials produced through our method. However, the wet chemical etching may potentially compromise both our material integrity and structural geometries^{7, 8}.

Therefore, to determine, verify, and cross-validate the results of Si, O, and C elemental ratios, we conducted SEM-EDS measurements using three different systems (ZEISS SUPRA 55, Phenom Pro, and FEI Quanta 450 FEG). Furthermore, we carried out decarbonization operations within the last equipment chamber (FEI Quanta 450 FEG) to mitigate the impact of carbon depositions.

The tests were carried out carefully in accordance with the comment. Once the data collection is finished, the electron beam moves rapidly to other areas to avoid surface carbon accumulation caused by prolonged focusing. The EDS spectra obtained from three equipment are provided below.

ZEISS SUPRA 55:

Phenom Pro:

FEI Quanta 450 FEG:

For all the measurement, the first groups (a, d & g) represent the commercial reference fused silica glass; the second groups (b, e & h) represent the surface of the printed fused silica glass; while the third groups (c, f & i) represent the cross-section of the printed fused silica glass, which is located 250 μm below the surface of the printed fused silica glass. The analysis of all EDS test data revealed that there was a high degree of consistency and marginal differences between the printed fused silica glass and the commercial conference samples.

Therefore, all 4 groups of SEM-EDS results have been updated in the Supplementary Data Fig. S14, with a table included in the Supplementary Data Table S1. Moreover, the sentence in manuscript is also updated in Page 14-15 to enhance comprehension. The corresponding texts are highlighted in red. "... A high level of consistency in binding energy, intensity, and atomic percentages of all elements were detected within the three groups (Supplementary Data Fig. S14 a-l & Table S1)."

“

1st (ZEISS SUPRA 55):

2nd (ZEISS SUPRA 55):

3rd (Phenom Pro):

4th (FEI Quanta 450 FEG):

Supplementary Data Fig. S14. SEM-EDS spectra of the (a, d, g & j) commercial and O_μSL 3D printed fused silica glass ((b, e, h & k) surface & (c, f, i & l) cross-section).”

“Supplementary Data Table S1 EDS results of the commercial and printed fused silica glass monoliths

Atomic percent, at. %												
Element	Commercial				Printed				Cross-section (Printed)			
	1 st	2 nd	3 rd	4 th	1 st	2 nd	3 rd	4 th	1 st	2 nd	3 rd	4 th
C	9.74	9.74	10.25	7.02	9.71	9.40	10.25	6.89	9.74	9.93	10.54	7.17
O	62.64	62.60	62.11	64.72	62.44	62.67	62.11	63.56	61.73	62.45	60.41	62.55
Si	27.62	27.66	27.65	28.26	27.85	27.93	27.65	29.55	28.54	27.62	29.05	30.28
Total	100											

”

XPS:

6. The broad spectrum in Fig. S14 a contains many unidentified peaks. The authors must identify and label the peaks visible in the spectrum.

Response: Your observation is appreciated. All the peaks in the full X-ray photoelectron spectroscopy (XPS) analysis (Supplementary Data Fig. S15) have been identified and labeled, and an offset has been also conducted to facilitate the comparison between the printed and the reference samples. The corresponding results are updated in Page 15 and Supplementary Data Fig. S15, with texts highlighted in red.

“... The printed fused silica glass is composed of pure Si and O element, as confirmed by the full X-ray photoelectron spectroscopy (XPS) analysis (0 - 1350 eV, Supplementary Data Fig. S15a), which aligns with the characteristics observed in the commercially available reference sample.

Supplementary Data Fig. S15. Characterization of printed-sintered fused silica glass compared with its commercial counterpart: (a) full XPS spectrum and fine XPS spectra of each element ...”

7. The authors must clarify how the energy axis was calibrated in the measurement, as this is a challenge for XPS of electrically insulating samples: see [10.1002/anie.201916000](https://doi.org/10.1002/anie.201916000)

Response: Thank you for the guidance provided. Since both commercial and printed fused silica glass are insulating samples, the adventitious carbon (C1s chemical state, C-C bond with a binding energy of 284.8 eV) was used for calibration in the XPS measurement. A sentence is added in Page 15 and corresponding text is highlighted in red. “The adventitious carbon (C1s state, C-C bond with a binding energy of 284.8 eV) was employed for XPS calibration, which is an indispensable procedure prior to the analysis of electrically insulating fused silica glass.”

8. The XPS data curves of the printed and the reference samples should be offset in

the y-axis direction to facilitate a comparison of the peak heights (as plotted now, the data of the printed sample hides the data of the reference sample).

Response: Your valuable guidance is greatly appreciated. The offset has now been conducted to facilitate the comparison between the printed and the reference samples, and all the peaks detected in the full XPS analysis (Supplementary Data Fig. S15a) have been identified and labeled. Please refer to Response #6 for more details.

9. The authors still do not report stoichiometric ratios obtained from the XPS measurement. The authors must provide this data for the printed and the reference samples.

Response: We express our gratitude for the considerate recommendation. The XPS stoichiometric ratios of all detected elements in both commercial and printed fused silica glass have been provided in Supplementary Data Table S2, and a sentence is also added in Page 15 with corresponding texts highlighted in red. "... indicating their identical chemical state and structural composition (Supplementary Data Table S2)."

“Supplementary Data Table S2 The stoichiometric ratios derived from the XPS spectra of commercial and printed fused silica glass

	Atomic composition, at. %			
	O	Si	C	N
Commercial	60.35	31.24	7.30	1.11
Printed	62.86	31.15	5.10	0.89

”

10. The authors claim that the Si2p, O1s, and O2s peaks of the printed and reference sample “displayed comparable photoelectron binding energy and relative intensity.” This is, however, impossible for the reader to verify since they have normalized each peak separately. The same normalization must be applied to each sample’s peaks to facilitate comparison of relative peak heights.

Response: We appreciate the reviewer’s thorough evaluation and kind advice. In the

XPS spectral analysis, we applied same normalizations on the corresponding peaks. To enhance the comprehension, we now provide the original data (binding energy *versus* counts) of the Si2*p*, O1*s*, and O2*s* peaks as presented below.

And a sentence is added in Page 15 to claim that the same normalization is applied to each peak, with corresponding text highlighted in red. “... The same normalizations were applied to each characteristic peak (526 - 536 eV for O1*s*, 97 - 107 eV for Si2*p*, and 0 - 35 eV for O2*s*) of both the printed and commercial samples, the fine spectroscopy results (inset of Supplementary Data Fig. S15a) revealed that both the printed and commercial samples displayed comparable photoelectron binding energy and relative intensity.”

11. There seems to be an energy offset between O1*s* and O2*s* peaks of the printed and reference samples of about 0.1 eVs and 0.5 eVs, respectively. The authors must clarify if this is a significant difference.

Response: We thank you for the careful observation and kind suggestion. The binding energy of the same chemical state is a range (531.0 ± 0.5 eV for O1*s* peak, and 23.6 ± 0.5 eV for O2*s* peak) rather than a fixed value. The obtained results fall within acceptable limits. A sentence is now included in Page 15, which is also highlighted in red. “... The peaks for the printed fused silica glass closely matched commercially stoichiometric fused silica glass (531.0 ± 0.5 eV for O1*s* peak, 101.7 ± 0.5 eV for Si2*p* peak, and 23.6 ± 0.5 eV for O2*s* peak) ...”

XRD:

12. The authors have not responded to the subject of my criticism of the XRD data

presented in Supplementary Data Fig S10b in the original submission. In the revised submission, the authors simply present new XRD data in Supplementary Data Fig. S14b, commenting that the data has been subjected to “straight line subtraction and deconvolution.” These terms have no physical meaning in the analysis of XRD data. The authors must clarify what was done differently in the new measurement and analysis and explain why their result now differs from what they presented in the original submission.

Response: We appreciate your thoughtful suggestion and sincerely apologize for the ambiguity in our explanation. We now conclude that the broad peak shift in original XRD results is attributed to the rough surface of printed fused silica glass. Numerous scratches occur on the platform surface because of using the razor blade to remove the printed samples. During the curing process, the coarse platform would reverse its scratches to the printed sample (typically in the lower surface). The following SEM images illustrate the surface microstructures of the initially tested samples (both commercial and printed).

According to Bragg's law $n\lambda = 2d \sin \theta$, even if the diffraction order n , wavelength λ and “grating constant” of the crystal d remain constants, variations in the glancing angle θ would still occur due to irregularities (such as low flatness and high roughness) of the tested surface^{9, 10}. Please refer to the schematics provided below.

As a result, the entire peak of the initially tested printed sample shifted in the XRD spectrum due to its coarse surface. Please refer to the following XRD spectra.

Therefore, in our recent XRD experiment, we reprinted the fused silica glass monolith and performed polishing on the two sides. The printed fused silica glass sample is carefully positioned on the testing platform to ensure it is placed at the center. The SEM images illustrating the surface microstructures of the recently tested samples (both commercial and printed) are presented below.

Therefore, a high level of consistency can be achieved in XRD; please refer to the spectra.

A sentence is added in Page 16 with corresponding references updated. And the text is highlighted in red. "... In this test, the printed fused silica glass monolith was carefully polished on both sides to eliminate the influence of surface roughness^{42, 43}."

TEM:

13. The authors have performed TEM and EDS measurements of a cross-section of a printed sample. However, the authors only show the Si, O, and Pt distribution without any quantification. Also, the primary possible contaminant, carbon (C), is left out of the analysis. The authors should provide the quantification of the Si/O stoichiometry and the amount of carbon measured.

Response: We appreciate your kind advice. The distribution of the C element has been incorporated, along with the quantification of all stoichiometry elements in the TEM results. Corresponding images are updated in Supplementary Data Fig. S17 to enhance comprehension.

"The transmission electron microscopy (TEM) images in Supplementary Data Fig. S17a & b demonstrate the dense nature of the printed fused silica glass structures, exhibiting an absence of discernible pores or cracks. The Supplementary Data Fig. S17c-e presents energy-dispersive spectroscopy mapping, which reveals the homogeneous distributions of Si and O elements. The diffraction patterns (inset of Supplementary Data Fig. S17b) corresponding to the amorphous phase exhibit excellent agreement with the X-ray diffraction analysis.

Supplementary Data Fig. S17. **a** The printed fused silica glass after Focus Ion Beam (FIB) milling. **b** TEM image and electron diffraction pattern of the printed fused silica glass. EDS mappings of **(c)** total, **(d)** Si, **(e)** O, **(f)** Pt, and **(g)** C element.”

Similar to the Response #4, quantitative EDS analysis would always be compromised for the light elements like carbon^{1, 2, 3, 4, 5, 6} due to its inherent limitations and other contributing factors such as coating, substrate, vacuum degree, etc. A sentence about the result of TEM-EDS is added into Page 16&17 in the text and is highlighted in red. “... According to Supplementary Data Fig. S18, the atomic percentage (at %) of silicon was measured to be 33.8 ± 0.1 , while that of oxygen was found to be 63.4 ± 0.1 at %. These values closely corresponded to the stoichiometric SiO_2 .” And the corresponding EDS result is added as Supplementary Data Fig. S18. “

Supplementary Data Fig. S18. TEM-EDS spectrum of the O μ SL 3D printed fused silica glass.”

And the conclusive sentence in Page 17 is updated with corresponding text highlighted in red. "... Consequently, the integration of XPS, XRD, Raman spectrum, TEM, and electron diffraction (Supplementary Data Fig. S15, S17 & S18, Table S2 & S3) identifies the final materials to be comparable to the stoichiometrically pure silica."

While the sentence in Page 2 regarding silica glass printed from organosilicon resins is now revised. And the corresponding texts are highlighted in red. "... Even though it has been reported recently ^{20, 21, 22, 23} that organosilicon resins are capable of fabricating high-precision glass optics below the glass transition temperature, this technology is still in its early stage and only few appropriate organic precursors have been developed so far. The nanocomposite system is still widely considered a more sophisticated solution, and in principle, it is better suited for the fabrication of pure, high-quality fused silica glass for applications in microelectronics and micro-/nano- photonics."

Printing resolution:

- The authors write in their abstract:

"our study demonstrates ... that 3D printing based on one-photon micro-stereolithography (O μ SL) permits the flexible creation of transparent and high performance fused silica glass components with complex, 3D SUB-MICRON ARCHITECTURES". (emphasis added)

and

"The methodology facilitates the construction of fused silica glass components with ARBITRARY 3D GEOMETRIES FEATURING FINE DETAILS AS SMALL AS 0.8 μ M". (emphasis added)

14. In my opinion, the authors have not demonstrated this in their work.

Response: The original direct measurement results obtained from SEM analysis on the sub-micron features of both suspending thread and line array are now provided below for better understanding.

Sub-micron thread:

Sub-micron lines:

Sub-micron and gradually increasing spacings:

To further enhance comprehension, we have also demonstrated a fused silica glass hierarchical lattice structure with sub-micron struts as its minimum features. The corresponding SEM images are now presented in Fig. 3, while the statement is incorporated on Page 13 of the text and highlighted in red.

“Additionally, we demonstrate an O μ SL 3D-printed fused silica glass hierarchical lattice structure (Fig. 3), which exhibits feature size spanning nearly 5 orders of magnitude, ranging from millimetric scale (Fig. 3a) to sub-micron scale (Fig. 3h & Supplementary Data Fig. S11d). The multiscale structure establishes a hierarchical connection from millimetric architectures to sub-micron features, progressively reducing its feature size by several times at each level. Our proposed technique enables the creation of 3D fused silica glass with simultaneous macroscale architectures (Fig. 3a) and sub-micron-scale features (Fig. 3h & Supplementary Data Fig. S11d), a combination that has not been reported in previous 3D-printed fused silica glass^{11, 12, 16, 18, 19, 20, 21, 22, 23, 31, 32, 33, 34, 35} and can be barely achieved using other lithography-based 3D printing techniques^{16, 18, 19, 20, 21, 22, 23, 31, 33, 34}.

Fig. 3: O μ SL 3D-printed fused silica glass hierarchical lattice structure with multi-scale critical features. Optical microscope images of the O μ SL 3D-printed fused silica glass hierarchical (a) lattice structure, (b) lattice network and (c) unit cell. (d-h) Electron microscope images depicting features of structurally hierarchical fused silica glass lattice unit cell shown in (c) down to sub-micron in strut size.”

And the SEM direct measurement result of sub-micron strut is also provided below.

And the results are now presented in Supplementary Data Fig. S11a-d. “

Supplementary Data Fig. S11. The direct measurement results obtained from SEM analysis on the $O\mu$ SL 3D-printed fused silica glass sub-micron features of (a) line width, (b) line spacing, (c) thread, (d) strut ...”

15. Furthermore, they ignored my explicit request to clearly state in the abstract the dimensions of the smallest voxel that can be printed.

Response: We apologize sincerely and profoundly for our oversight. Now the statement regarding the smallest voxel fabricated using our method has been included in the abstract, with the relevant text highlighted in red. “... The incorporation of homogeneous volumetric shrinkage further facilitates the attainment of the smallest voxel sizes, reducing it from $2.0 \times 2.0 \times 1.0 \mu\text{m}^3$ to $0.8 \times 0.8 \times 0.5 \mu\text{m}^3$.” Please also refer to the following response (Response #16) for our detailed explanation of the printed layer thickness.

16. From supplementary Figure S5 b, we see that the printed layer thickness is about $2.5 \mu\text{m}$.

Response: The critical resolution of printed layer is co-determined by the lithography system and the photosensitive properties of precursor. According to the Beer-Lambert law, the theoretical equation for determining the thickness C_d of a single-layer polymerization can be derived as follows:

$$C_d = D_p \cdot (\ln E - \ln E_c) \quad (1)$$

where D_p is the penetration depth of the precursor resin, indicating the maximum distance that UV light can reach into the resin; E is the exposure energy input by the lithography system; while E_c is the critical exposure energy, representing an energy threshold that triggers the crosslinking of polymers.

As depicted in the Page 6, “... the photo-absorbers decrease the absorption coefficients of precursor (Supplementary Data Fig. S5a), preventing shape errors caused by overexposure and obtaining a uniform thickness of approximately 2 μm (Supplementary Data Fig. S5b) ...” Supplementary Data Fig. S5a presents a comparison of the resin penetration depth D_p before and after modulating the absorption coefficient using two carefully selected photo-absorbers. Therefore, the printed layer thickness (2.5 μm) in Supplementary Data Fig. S5b represents the penetration depth D_p (the maximum distance reached by UV light) of nanocomposite photo-polymerizable precursor resin after the modulation of absorption coefficient, rather than the minimum achievable layer thickness.

By precisely adjusting the input exposure energy and slice thickness (with a minimum of 1 µm for the printing system), it is possible to decrease the thickness of a printed fused silica glass monolayer to approximately 500 nm after post-processing, as demonstrated in the SEM image below. A sentence is added in Page 6 with highlighted red texts, and the corresponding SEM image is now presented in Supplementary Data Fig. S11e. “... By precisely adjusting the input exposure energy and slice thickness (with a minimum of 1 µm for the printing system), it is possible to further reduce the thickness of a printed fused silica glass monolayer to approximately 500 nm after post-processing (Supplementary Data Fig. S11e) ...”

Supplementary Data Fig. S11. The direct measurement results obtained from SEM analysis on the OµSL 3D-printed fused silica glass sub-micron features of ... (e) layer.”

17. Judging from the new supplementary Figure S6, the layers shrink to 1.9 μm upon decarbonization at 600 $^{\circ}\text{C}$. The layers then reach an ultimate average thickness of 0.9 μm after sintering at 1050 $^{\circ}\text{C}$, but the layer-to-layer variation in thickness is a few hundred of nanometres.

Response: We appreciate your meticulous observation and comprehensive assessment. As written in Page 7, “The silica backbones are converted into dense amorphous fused silica glass during the final sintering steps at 1050 $^{\circ}\text{C}$ under vacuum.” The silica nanoparticles undergo melting and joining during the high-temperature sintering process, which is benefit to the structural smoothing and surface roughness decreasing. The aforementioned process would lead to structural rounding and subsequently result in dimensional variation in the adjacent contour (layer thickness).

18. Furthermore, after the high-temperature sintering, a significant rounding of the structures is visible, further limiting the resolution. This rounding effect is also visible in the new Figure S10.

Response: We appreciate your careful observation. The structural rounding effect, on the other hand, significantly facilitates the enhancement of surface roughness in the final fused silica glass product. The surface fabricated by O μ SL 3D-printing technique consists of an array of 2 μm voxels with high surface roughness. While the achievement of a high-quality and defect-free surface with low surface roughness is attributed to the structural rounding effect accompanied by intricate stepwise post-processing procedures. The SEM images illustrating the printed surface before and after post-processing are now presented below to help understanding.

Before post-processing (high surface roughness):

After post-processing (low surface roughness):

As demonstrated in Page 18 and Fig. 4b in manuscript, the white light interferometer confirmed a desired surface roughness of $R_a \approx 0.633$ nm, thereby enhancing the optical capabilities of the O μ SL 3D-printed fused silica glass micro-optics.

19. Looking more closely at Figure S10, it is clear that the rounding after sintering is a severe limitation. It is thus unclear why the authors write in the caption of Figure S10 that it is “suggesting the sub-micron features in both line width and spacing,” I don’t observe anything resolved in the image with dimensions below one micron. In Figure S10 d, the smallest resolved line is 1.4 μm , and the smallest resolved gap is 1.2 μm .

Response: Thank you for your thorough evaluation. The direct measurement results obtained from SEM analysis on the sub-micron features are provided in Response #14. Please kindly refer to it for detailed information on sub-micron line width and spacing.

20. Furthermore, the authors failed to state what line and gap design they aimed to print—

Response: Thank you for your kind advice. Now, a sentence regarding the design we aim to print has been added to Page 8, with the relevant texts highlighted in red. “The dimensions of both line width and spacing successively increase from 2 μm (equivalent to one pixel) to 8 μm (equivalent to four pixels), respectively.”

21. Finally, the patterns in Figs. S6 and S10 don’t contain any suspended lines. Without that, one cannot state anything about the resolution of a 3D printing process that allows the creation of “arbitrary 3d geometries,” in the author’s own words.

Response: Your thoughtful evaluation and suggestion are greatly appreciated. We have demonstrated several types of sub-micron features fabricated using the proposed O μ SL

3D-printing technology, please kindly refer to Response #14 for details. All the words previously labeled as “arbitrary” have now been revised to “miscellaneous”, and the corresponding terms are now highlighted in red.

22. From my viewpoint, the authors cherry-pick the smallest obtained values in one dimension without stating the values for the two others at the same conditions. Judging from the limited information provided, I estimate that this method can print a voxel of size about $2 \times 2 \times 3 \mu\text{m}^3$ before post-processing, which shrinks to about $1 \times 1 \times 1 \mu\text{m}^3$ after post-processing. Any claims of sub-micron patterning capability remain unfounded.

Response: We express our gratitude for the reviewer's comprehensive assessment and meticulous examination. The direct measurement results obtained from SEM analysis on the sub-micron features are provided in Response #14 (for XY-dimension) and Response #16 (for Z-direction). Please kindly refer to them for detailed information. Also, the declaration on the printed voxel size before and after post-processing is now included in the abstract, and the corresponding text is highlighted in red. “... The incorporation of homogeneous volumetric shrinkage further facilitates the attainment of the smallest voxel sizes, reducing it from $2.0 \times 2.0 \times 1.0 \mu\text{m}^3$ to $0.8 \times 0.8 \times 0.5 \mu\text{m}^3$.”

23. The placement of the white scale bar on the bright (charging) area in Fig. S10 d makes it very difficult to see. Use a black scale bar or move the white scale bar to a dark background.

Response: Thank you for your suggestion. Now the scale bar is updated with black color, please refer to the Supplementary Data Fig. S10 for more details.

“

Supplementary Data Fig. S10. (a) Optical & (b-d) electron microscopic images of a 15×10 column array affixed line arrays on the tops, suggesting the sub-micron features in both line width and spacing.”

24. The authors must state CLEARLY in their abstract the three dimensions of the smallest resolved voxel BEFORE and AFTER post-processing. The obligatory high-temperature sintering is a severe limitation for printing on a substrate. Only very few substrates survive heating to 1050 °C, and the shrinkage of the printed structure with respect to the substrate will cause deformation and even cracking and detachment. Without this information clearly stated in the abstract, it is impossible to compare the advance of this work to previous similar work, some of which require no post-processing.

Response: Appreciation for your thoughtful recommendation. The declaration on the printed voxel size before and after post-processing is now included in the abstract, please refer to Response #15 and Response #22. The proposed technique allows for the construction of both micro-/nano- features and macro-architecture. The macro structures enable us to remove the printed parts from the substrate before heat treatments. Please refer to the procedure of removal in *Method - OuSL process*

(Page 29) of the manuscript, "... Finally, the as-dried parts were removed from the substrate with a razor blade, in which great care must be taken as the microstructure is delicate and easily broken." Such removal effectively prevents the deformation and defects caused by adhesion, benefiting the achievement of high-quality fused silica glass components.

25. Since the printing approach applied here is based on linear absorption of light at a wavelength of 405 nm, it would be highly unexpected to achieve sub-micron resolution with this technique, and indeed, the results presented do not support such a claim. The only 3D printing methods convincingly demonstrating sub-micron resolution are based on non-linear (multiphoton) absorption.

Response: In our experiments, the 800 nm resolution is produced by high shrinkage of the resins cured using a 2 μm -capable instrument. However, the potential resolution by one-photon lithography can exceed this value. As shown in the figures below, our precursor can be fabricated with intrinsic sub-micron features using a higher-resolution one-photon lithography instrument (also with the same working wavelength at 405 nm) prior to the post-processing.

Reviewer #2 (Remarks to the Author):

The revision is thorough and well executed, compliments to the authors, im happy to recommend for publication.

Response: The valuable advice and insightful suggestions you have provided greatly contribute to the enhancement of the scientific basis of the manuscript. We sincerely express our appreciation.

Reference

1. Thomas LE. Light-Element Analysis with Electrons and X-Rays in a High-Resolution Stem. *Ultramicroscopy* **18**, 173-184 (1985).
2. Goldstein JI, Newbury DE, Echlin P, Joy DC. *Scanning Electron Microscopy and X-Ray Microanalysis: A Text for Biologists, Materials Scientists, and Geologists*, second edn. Springer (1992).
3. Heu R, Shahbazmohamadi S, Yorston J, Capeder P. Target Material Selection for Sputter Coating of SEM Samples. *Microscopy Today* **27**, 32-36 (2019).
4. Ramalingam B, Mukherjee S, Mathai CJ, Gangopadhyay K, Gangopadhyay S. Sub-2 nm size and density tunable platinum nanoparticles using room temperature tilted-target sputtering. *Nanotechnology* **24**, 205602 (2013).
5. Takeuchi A, Wise H. High Dispersion Platinum Catalyst by Rf Sputtering. *J Catal* **83**, 477-479 (1983).
6. Yoshida A, Kaburagi Y, Hishiyama Y. Chapter 5 - Scanning Electron Microscopy. *Materials Science and Engineering of Carbon*, 71-93 (2016).
7. Hülsenberg D, Harnisch A, Bismarck A. *Microstructuring of Glasses*. Springer Berlin Heidelberg, 2008 (2005).
8. Van Toan N, Toda M, Ono T. An Investigation of Processes for Glass Micromachining. *Micromachines (Basel)* **7**, (2016).
9. Pitschke W, Hermann H, Mattern N. The influence of surface roughness on diffracted X-ray intensities in Bragg–Brentano geometry and its effect on the structure determination by means of Rietveld analysis. *Powder Diffraction* **8**, 74-83 (1993).

10. Moram MA, Johnston CF, Kappers MJ, Humphreys CJ. The effects of film surface roughness on x-ray diffraction of nonpolar gallium nitride films. *J Phys D Appl Phys* **42**, (2009).

REVIEWERS' COMMENTS

Reviewer #1 (Remarks to the Author):

The authors have resolved well all my questions regarding the manuscript.

Reviewers' comments:

Reviewer #1 (Remarks to the Author):

The authors have resolved well all my questions regarding the manuscript.

Response: The valuable advice and insightful suggestions you have provided greatly contribute to the enhancement of the scientific basis of the manuscript. We sincerely express our appreciation.